# All-optical closed-loop voltage clamp for precise control of muscles and neurons in live animals

Amelie C. F. Bergs [1,2], Jana F. Liewald [1,2], Silvia Rodriguez-Rozada[3], Qiang Liu [4,5], Christin Wirt[1,2], Artur Bessel[6], Nadja Zeitzschel [1,2], Hilal Durmaz[1,2], Adrianna Nozownik[3], Holger Dill[1,2], Maëlle Jospin[7], Johannes Vierock [8], Cornelia I. Bargmann[4,9], Peter Hegemann [8], J. Simon Wiegert [3,10] & Alexander Gottschalk [1,2] ✉

Excitable cells can be stimulated or inhibited by optogenetics. Since optogenetic actuation regimes are often static, neurons and circuits can quickly adapt, allowing perturbation, but not true control. Hence, we established an optogenetic voltage-clamp (OVC). The voltage-indicator QuasAr2 provides information for fast, closed-loop optical feedback to the bidirectional optogenetic actuator BiPOLES. Voltage-dependent fluorescence is held within tight margins, thus clamping the cell to distinct potentials. We established the OVC in muscles and neurons of *Caenorhabditis elegans*, and transferred it to rat hippocampal neurons in slice culture. Fluorescence signals were calibrated to electrically measured potentials, and wavelengths to currents, enabling to determine optical I/V-relationships. The OVC reports on homeostatically altered cellular physiology in mutants and on $Ca^{2+}$-channel properties, and can dynamically clamp spiking in *C. elegans*. Combining non-invasive imaging with control capabilities of electrophysiology, the OVC facilitates high-throughput, contact-less electrophysiology in individual cells and paves the way for true optogenetic control in behaving animals.

Identifying connections between distinct neurons and their contribution to driving behavior is a central issue in neuroscience[1,2]. To explore such relations, methods to record and concurrently regulate neural activity are needed[3]. Also, high-throughput screening approaches in excitable cell physiology require application of such methods. Diverse approaches to control or observe excitable cell function are in use. Patch-clamp electrophysiology provides superior temporal accuracy

and sensitivity, but is limited by its invasiveness[4–6]. $Ca^{2+}$ imaging, applicable in intact living organisms, enables integrating cell physiology and behavioral output[7–9]. However, due to sensor $Ca^{2+}$ buffering and the non-linear correlation of cytosolic $Ca^{2+}$ concentration and membrane voltage, this technique suffers from comparably low temporal resolution, and fails to resolve subthreshold voltage transients or high-frequency action potentials (APs). Further, since $Ca^{2+}$

[1]Buchmann Institute for Molecular Life Sciences, Goethe University, Max-von-Laue-Strasse 15, 60438 Frankfurt, Germany. [2]Institute of Biophysical Chemistry, Goethe University, Max-von-Laue-Strasse 9, 60438 Frankfurt, Germany. [3]Research Group Synaptic Wiring and Information Processing, Center for Molecular Neurobiology Hamburg, University Medical Center Hamburg-Eppendorf, 20251 Hamburg, Germany. [4]Lulu and Anthony Wang Laboratory of Neural Circuits and Behavior, The Rockefeller University, New York, NY 10065, USA. [5]Department of Neuroscience, City University of Hong Kong, Tat Chee Avenue, Kowloon Tong, Hong Kong, China. [6]Independent Researcher, Melatener Strasse 93, 52074 Aachen, Germany. [7]Université Claude Bernard Lyon 1, Institut Neuro-MyoGène, 8 Avenue Rockefeller, 69008 Lyon, France. [8]Institute for Biology, Experimental Biophysics, Humboldt University, 10115 Berlin, Germany. [9]Chan Zuckerberg Initiative, Palo Alto, CA, USA. [10]Medical Faculty Mannheim, University of Heidelberg, Ludolf-Krehl-Strasse 13-17, 68167 Mannheim, Germany. ✉e-mail: a.gottschalk@em.uni-frankfurt.de

concentration does not fall below basal cytosolic levels in most neurons, $Ca^{2+}$ imaging is unsuited to reveal synaptic inhibition. Instead, membrane potential can be imaged via genetically encoded voltage indicators (GEVIs), e.g., rhodopsin-based GEVIs, among others[10–17]. The fluorescence of retinal, embedded in rhodopsins, reliably monitors voltage dynamics at millisecond timescales. Since rhodopsin-based GEVIs emit near-infrared light, they can be multiplexed with optogenetic actuators of membrane currents, to selectively photostimulate or inhibit activity of excitable cells with high spatiotemporal precision[18–20]. In the "(i)-Optopatch" approach, the blue-light activated channelrhodopsin (ChR) CheRiff was used together with the voltage indicator QuasAr2, allowing to unidirectionally steer and observe, but not to fully control neuronal activity[11,12,21]. An optical dynamic clamp (ODC) used archaerhodopsin to mimic $K^+$ currents, absent in immature cardiomyocytes, in closed loop with electrophysiological feedback[22]. Bidirectional optical modulation and readout of voltage was implemented as light-induced electrophysiology (LiEp) for drug screening, however, without a feedback loop[23]. A bidirectional approach with feedback—"optoclamp"—used ChR2 and halorhodopsin (NpHR) as actuators and extracellular microelectrode arrays instead of GEVIs[24]. This approach clamped average firing rates in neuronal ensembles, following indirect voltage readout.

True all-optical control over excitable cell activity with closed-loop feedback, as in voltage-clamp electrophysiology, should combine two opposing optogenetic actuators for de- and hyperpolarization, as well as a GEVI. Such a system could respond to intrinsic changes in membrane potential. It would also prevent inconsistent activity levels arising from cell-to-cell variation in optogenetic tool expression levels. These are usually not taken into account, particularly when static stimulation patterns are used, while a feedback system could increase stimulation until the desired activity is reached. The system should synergize the non-invasive character of imaging methods with the control capabilities of electrophysiology.

Here, we established an optogenetic voltage-clamp (OVC) in *Caenorhabditis elegans* and explored its use in rat hippocampal organotypic slices. The OVC uses QuasAr2 for voltage read-out[11,20], and BiPOLES, a tandem protein comprising the depolarizer Chrimson and the hyperpolarizer *Gt*ACR2, stimulated by orange and blue light, respectively, for actuation[19,25–28]. Spectral separation and their balanced 1:1 expression in BiPOLES enabled gradual transitions from depolarized to hyperpolarized states, and vice versa. QuasAr2 fluorescence was sampled at a rate of up to 100 Hz, and this information was used to compute a feedback of wavelength-adapted light signals transmitted to BiPOLES, in closed-loop, thus keeping the voltage-dependent fluorescence at a desired level. We characterized the system in body-wall muscle cells (BWMs), as well as in cholinergic and GABAergic motor neurons. Simultaneous measurements allowed calibrating fluorescent signals to actual membrane voltages, and passively presented wavelength pulses to currents. In *unc-13* mutants, the OVC readily detected altered excitability of muscle, as a response to the reduced presynaptic input. In *egl-19* VGCC gain-of-function (g.o.f.) mutants, an optical I/V-relationship of the mutated channel could be obtained, comparing well to electrophysiological measurements. In rodent neurons, the OVC also modulated fluorescence and voltage, yet with a smaller range, due to the resting potential being close to $Cl^-$ reversal potential. Last, in spontaneously active tissues, i.e., pharyngeal muscle and the motor neuron DVB[29–32], the OVC could dynamically follow and counteract native APs, and suppress associated behaviors.

## Results

### Reading GEVI fluorescence to steer optogenetic actuators with light feedback in closed-loop

An OVC should measure voltage-dependent fluorescence of an excitable cell, e.g., via a rhodopsin-GEVI[10,11], and provide adjusted light-feedback to optogenetic actuators of membrane voltage, e.g.,

cation- and anion-selective rhodopsin channels[19,28,33,34] (Fig. 1A). The OVC needs to work in closed-loop, to quickly counteract intrinsic activity. We first implemented suitable hard- and software (Fig. 1B and Supplementary Code 1): To excite fluorescence of rhodopsin GEVIs (e.g., QuasAr2, emitting in the far-red)[10,11], we expanded a 637 nm laser to cover 0.025 mm². GEVI fluorescence of a region of interest (ROI) is monitored by a camera, and compared to a target value (Fig. 1C and Supplementary Fig. 1A–C). Light feedback is sent to the sample from a monochromator, whose wavelength limits can be pre-selected to match the chosen hyperpolarizing and depolarizing optogenetic tools' maximal activation, and that can adjust wavelength at 100 µs and 0.1 nm temporal and spectral resolution, respectively.

Communication between camera and monochromator is provided by a custom-written script in Beanshell (part of µManager interface[35]), processing incoming gray values into relative changes of fluorescence ($\Delta F/F_0$) (Fig. 1C, Supplementary Fig. 1A–C, and Supplementary Code 1). We initially performed step response experiments in open loop configuration and found the system to require barely 20 ms to reach the desired OVC step-input of ±5% $\Delta F/F_0$ (see below), which we conclude to approximate the system's time constant (Supplementary Fig. 1D). Due to excitation of QuasAr with a laser, 10 ms exposure sufficed to collect ca. 1,800,000 photons per ROI per frame. Hence, the shot-noise floor is as low as ca. 0.08% of the total signal power (Supplementary Table 1). Since GEVIs exhibit photobleaching, $\Delta F/F_0$ values would gradually deviate from actual voltage levels. Thus, for each recording, an initial calibration phase is used to calculate correction parameters (Supplementary Fig. 1E, F). Once the system can access bleach-corrected $\Delta F/F_0$ values, it feeds them into a decision tree algorithm (Supplementary Fig. 1A), where they are compared to a desired holding $\Delta F/F_0$ value. Deviation between target and actual $\Delta F/F_0$ determines the wavelength change of the monochromator (I(ntegral)-controller). To increase control stability, a selectable tolerance range was defined (in most cases ±1%), in which the actual value is allowed to fluctuate around the target $\Delta F/F_0$. Once this tolerance range is reached, the control variable wavelength is not further changed. An alternative algorithm, using a PID controller[36] and Kalman filter[37] for sensor smoothing, did not increase overall system performance (see "Methods"; Supplementary Fig. 2).

### Combining single actuators with QuasAr2 for unidirectional steering of membrane voltage

First, we expressed QuasAr2 in *C. elegans* body wall muscles (BWMs). Expression levels were very uniform across and within strains, ensuring equal OVC activity in different genetic backgrounds (Supplementary Fig. 1G). 20 s calibration under 637 nm laser light sufficed to estimate photobleaching parameters. 300 µW/mm² blue light or presenting the full spectrum (400 to 600 nm) caused no additional bleaching and did not affect QuasAr2 fluorescence (Supplementary Fig. 1E, F, H). Thus, monochromator light did not influence bleaching-corrected fluorescence and estimated membrane voltages. Initially, to test the feedback loop, we assessed different actuators and configurations. First, *Chlamydomonas reinhardtii* ChR2(H134R) or *Guillardia theta* anion channelrhodopsin *Gt*ACR2 were expressed in cholinergic motor neurons (Supplementary Fig. 3A). This allowed manipulating muscle voltage indirectly via light-induced (de-)activation of motor neurons, and to adjust QuasAr2 fluorescence to values between +20 and −15% $\Delta F/F_0$, respectively (Supplementary Fig. 3B–D). However, fluorescence (i.e., voltage) returned to baseline only by intrinsic membrane potential relaxation (Supplementary Fig. 3D–G), thus limiting temporal resolution. The time required to reach the tolerance range of the target $\Delta F/F_0$ value was termed "transition time". For ChR2, fluorescence reached the target range after $150 \pm 12.7$ ms and relaxed within $678 \pm 148.9$ ms ($254 \pm 20.4$ ms and $239.5 \pm 29.9$ ms, respectively, for *Gt*ACR2). Next, we

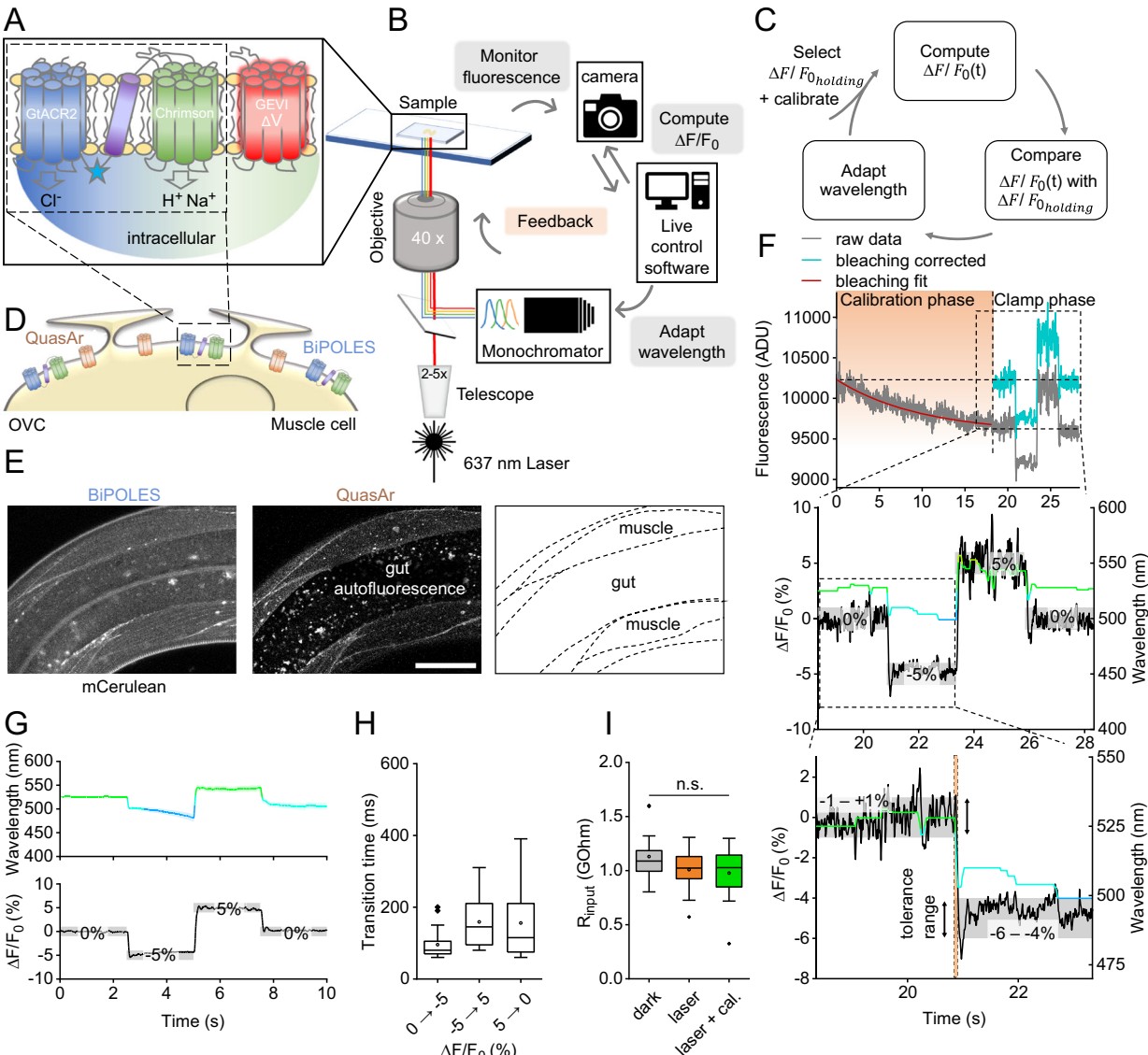

**Fig. 1 | Components, setup, and functionality of the OVC. A** Molecular OVC components: Two counteracting optogenetic actuators for de- and hyperpolarization (GtACR2, Chrimson; blue star: mCerulean), and a voltage indicator. **B** Hardware setup and communication: Membrane voltage is monitored via a fast and sensitive (sCMOS or EMCCD) camera; gray values are processed by live control software (Supplementary Code 1). Based on the difference to the set holding value, a feedback of light from a monochromator is sent to the optogenetic actuators. **C** OVC feedback mechanism: After selection of clamping parameters and calibration for bleaching correction, $\Delta F/F_0$ is calculated and compared with the set value, to adapt the wavelength accordingly, in closed-loop. **D** OVC (QuasAr2 and BiPOLES) in BWMs. **E** Confocal fluorescence $z$-projection. BiPOLES (mCerulean) and QuasAr2 in BWM membranes, scheme (dashed lines represent muscle plasma membranes). Scale bar, 50 μm. Representative image (from $n = 22$). **F** OVC four-step protocol

(0, −5, 5, and again 0% $\Delta F/F_0$) in BWMs; insets and below: close-up. Wavelength shown in the respective color, holding values and tolerance range (gray boxes) are indicated for each step. Orange shade in lower panel: transition period to reach tolerance range. ADU: analog-to-digital units. **G** Upper panel: Overlay of mean (±S.E.M.) wavelength and (lower panel) fluorescence traces ($n = 24$; holding values: 0, −5, 5, 0% $\Delta F/F_0$). **H** Times required for the indicated 5 and 10% $\Delta F/F_0$ transitions ($n = 24$). **I** Membrane resistance of BWM cells, before ($n = 13$), and during 637 nm laser illumination (orange, $n = 13$), and during laser + calibration light (green, $n = 12$). One-way ANOVA with Bonferroni correction (laser/dark: $p = 0.65$; laser+cal./dark: $p = 0.39$; laser+cal./laser: $p = 1$). In **H**, **I**, box plots (median, 25th–75th quartiles); open dot: mean; whiskers: 1.5× IQR. $n$−numbers refer to independent biological replicates (animals) (**G**–**I**). Source data are provided as a Source data file.

assessed combinations of spectrally distinct, opposing actuator pairs, like ChR2 (470 nm) and *Gt*ACR1 (515 nm), or *Natronomonas pharaonis* halorhodopsin NpHR (590 nm). Optogenetic de- or hyperpolarization of cholinergic motor neurons affects muscle activation and evokes body contraction or elongation[33,38] (Supplementary Fig. 4A–D). Yet, one actuator was typically outperformed by the other, impeding precise control of membrane potential. Likely, separate expression led to variable relative amounts of the tools. We thus resorted to 1:1 expression via BiPOLES.

## BiPOLES enables bidirectional voltage-clamping in *C. elegans* muscle

The tandem protein BiPOLES combines de- and hyperpolarizers Chrimson and GtACR2 (590 and 470 nm, respectively), linked as one sequence[27] (Fig. 1A). BiPOLES activation with a 400–600 nm ramp evoked robust bidirectional effects on body length (Supplementary Fig. 4E). We co-expressed BiPOLES and QuasAr2 in BWMs (Fig. 1D, E). Low levels of BiPOLES were found at the plasma membrane (PM) and in few intracellular aggregates, while QuasAr2 localized mainly to the PM.

As QuasAr2 excitation (637 nm) causes some activation of Chrimson (their spectra overlap[26,39]; Supplementary Fig. 5A), we needed to counteract its effects via GtACR2, by compensatory light from the monochromator. The wavelength was adjusted until −5 and 5% $\Delta F/F_0$ could be maintained for ca. 5 s. Additional assays ensured that compensatory light reinstates normal function: In animals expressing the OVC in BWMs or cholinergic neurons, 637 nm laser light diminished motor behavior. However, compensatory light restored body length and swimming behavior (Supplementary Fig. 5B–E and Supplementary Movie 1), and BWMs expressing the OVC showed voltage fluctuations comparable to animals expressing QuasAr2 only (Supplementary Fig. 5F–H). 637 nm laser light did not fully activate Chrimson, as it could be further excited by 590 nm (300 μW/mm²) light, increasing QuasAr2 fluorescence by 2.3% (Supplementary Fig. 5I). After bleaching correction, prior to each individual experiment ($R^2$ of the exponential fit was always >0.8, in most cases even >0.95; Supplementary Fig. 6A), the OVC generated incremental changes in wavelength that closely followed the fluctuating fluorescence signals (i.e., membrane voltage; Fig. 1F), as soon as the tolerance range for holding $\Delta F/F_0$ was exceeded. Cells were reliably and quickly (Supplementary Fig. 6B, C), clamped to holding values between −5 and 5% $\Delta F/F_0$ (Fig. 1F, G; Supplementary Fig. 6D). Due to the bidirectionality and live feedback, the BiPOLES-OVC acted significantly faster than when using single actuators, particularly for the return towards resting potential: Transition times were only 147.7 ± 25.5 ms (5→0%) and 88.0 ± 5.7 ms (−5→0%) for BiPOLES, compared to 678.6 ± 148.9 ms for ChR2 and 239.5 ± 29.9 ms for *Gt*ACR2 (Fig. 1H and Supplementary Figs. 3G and 6C–E). The transition time depended on the system's sampling rate: At 40 Hz sampling rate, the transition times were significantly larger, since the I-controller then applies fewer control signals to the system within a given period (Supplementary Fig. 6E). Evaluation of all control events (of 11 randomly selected experiments) demonstrated that almost 50% of those events were completed (i.e., $\Delta F/F_0$ back in tolerance) within 20-30 ms. These events where associated with smaller control deviations (Supplementary Fig. 6F). Analysis of the system's accuracy showed that the control deviation was within ±1% (corresponding to the selected tolerance range) at 84% of all time points, and 50% occurred even within ±0.5% (Supplementary Fig. 6G). An analysis of the time-dependent accuracy of the control error (root mean square of the deviation; r.m.s.d.) confirmed stability, apart from the periods of the transition time (Supplementary Fig. 6H). The OVC allowed continuous bidirectional clamping for extended periods (Supplementary Fig. 6I–L). Once the closed-loop control was interrupted, membrane voltage approached baseline and higher fluorescence fluctuations were observed (1.87 ± 0.1 vs. 0.78 ± 0.05% $\Delta F/F_0$ during −5% clamping; 1.89 ± 0.14% $\Delta F/F_0$ for animals expressing only QuasAr; Supplementary Fig. 5F–H). Importantly, the cell could also be actively steered back to the initial fluorescence level by the OVC (Fig. 1F, G).

## Calibrating QuasAr2 fluorescence and membrane potential in BWMs

To calibrate the OVC system and determine the actually accessible voltage range, we measured voltage and fluorescence simultaneously (Supplementary Fig. 7A). Concurrent laser (637 nm) and compensation illumination (520 nm) did not significantly alter membrane resistance (Fig. 1I), or APs, that could be observed by fluorescence and patch-clamp simultaneously (Fig. 2A and Supplementary Fig. 7B, C). The dual illumination also did not alter membrane potential; however, adding 470 nm light could hyperpolarize the cell by about 16 mV (Supplementary Fig. 7D). Small voltage fluctuations, likely representing EPSPs, were similar in dark and light conditions by amplitude and frequency (Supplementary Fig. 7E–G). Thus, BiPOLES activation, despite the open channels, did not lead to general shunting of membrane potential in *C. elegans* muscle. In simultaneous electrophysiology and optical OVC

experiments, induced fluorescence changes (−3 to +3% $\Delta F/F_0$, and returning back to 0% $\Delta F/F_0$) modulated voltage by ca. −7 to +8 mV (Fig. 2B–E), i.e., ca. 15 mV range per 6% $\Delta F/F_0$ (24 mV per 10% $\Delta F/F_0$; given the linear fluorescence-voltage relation of QuasAr2; ref. 11). Note, a smaller $\Delta F/F_0$ range was chosen to facilitate patch-clamp measurements; briefer optical clamp periods (400 ms instead of 2 s) allowed accessing broader voltage ranges (see below). We verified the fidelity of the OVC by assessing whether the initially deduced calibration function may cause progressive errors during the voltage clamp phase. Correlating induced $\Delta F/F_0$ traces to evoked membrane potentials revealed no (increasing) deviation of actual voltages from OVC-imposed trajectories (Supplementary Fig. 7H–K).

## Measuring all-optical I/V relationships

The OVC allowed reliable optical voltage clamping. We wondered if we could also use it to acquire all-optical I/V relationships. To this end, we devised a different software "pseudo I/V curve" (Supplementary Fig. 8A and Supplementary Code 1), which can consecutively present different $\Delta F/F_0$ clamp steps. First, we performed optical experiments, relating different $\Delta F/F_0$ clamp values (= voltage equivalent) to observed wavelengths (= current equivalent), required to achieve the respective fluorescence steps (Fig. 2F, G). These purely optical experiments (in intact animals) showed that a range of at least ±10% $\Delta F/F_0$ can be achieved with wavelengths of ca. 470–585 nm, different to the simultaneous OVC/patch-clamp experiments (Fig. 2H and Supplementary Fig. 8B). As an inverse control, we presented wavelengths and achieved a congruent output $\Delta F/F_0$ level (Supplementary Fig. 8C, D). Next, we calibrated the $\Delta F/F_0$ steps to actual voltages, and the wavelengths to actual currents. We again carried out simultaneous OVC/patch-clamp experiments and examined membrane voltage as a function of the pre-set clamp fluorescence (Fig. 2I). Individual steps were shortened to 400 ms, which allowed covering a range of ±5% $\Delta F/F_0$. The voltages determined and the range covered corresponded to those of the ±3% $\Delta F/F_0$ measurements using the 'standard' OVC protocol, i.e., ca. 22 mV (−40 to −18 mV) for ±5% $\Delta F/F_0$ (Fig. 2J). Likewise, we determined membrane currents (at −24 mV, i.e., BWM resting potential), induced by step changes in applied wavelengths (Fig. 2K). BiPOLES mediated currents in a total range of ~190 pA, running almost linearly between 420 and 580 nm (Fig. 2L). Linear regression estimations allowed relating membrane voltages to $\Delta F/F_0$ and currents to wavelengths, and to identify calibration parameters (Fig. 2J, L; "Methods"). Significance levels for the coefficients were <0.001, indicating high precision. Thus, averaged optical data can be converted to voltage and currents.

## Demonstrating homeostatic changes in muscle excitability using the OVC

We explored the utility of the OVC to assess divergent cell physiology. *unc-13(n2813)* mutants lack an essential synaptic vesicle priming factor[40] and thus exhibit largely reduced postsynaptic currents upon ChR2-stimulation of motor neurons[41]. Yet, *unc-13* and other neurotransmission mutants showed enhanced muscle contraction when BWMs were directly photostimulated. We hypothesized that muscles homeostatically change their excitability. To explore if the OVC can detect this, we induced a +5% $\Delta F/F_0$ depolarization step: In *unc-13(n2813)* mutants, the OVC required significantly blue-shifted light (i.e., less Chrimson activation) to induce the same level of depolarization as in wild type (534.2 ± 2.8 vs. 550.4 ± 3.1 nm; Fig. 3A, B). Muscles in *unc-13* animals may exhibit higher excitability to balance the lower excitatory input they receive. Though no significant change in membrane resistance was observed (Fig. 3C), the amplitude of induced voltage increases was higher in *unc-13* mutants when we injected current ramps (Fig. 3D–F). Similarly, APs measured by voltage imaging were significantly increased by amplitude and duration (Fig. 3G, H, J). Thus,

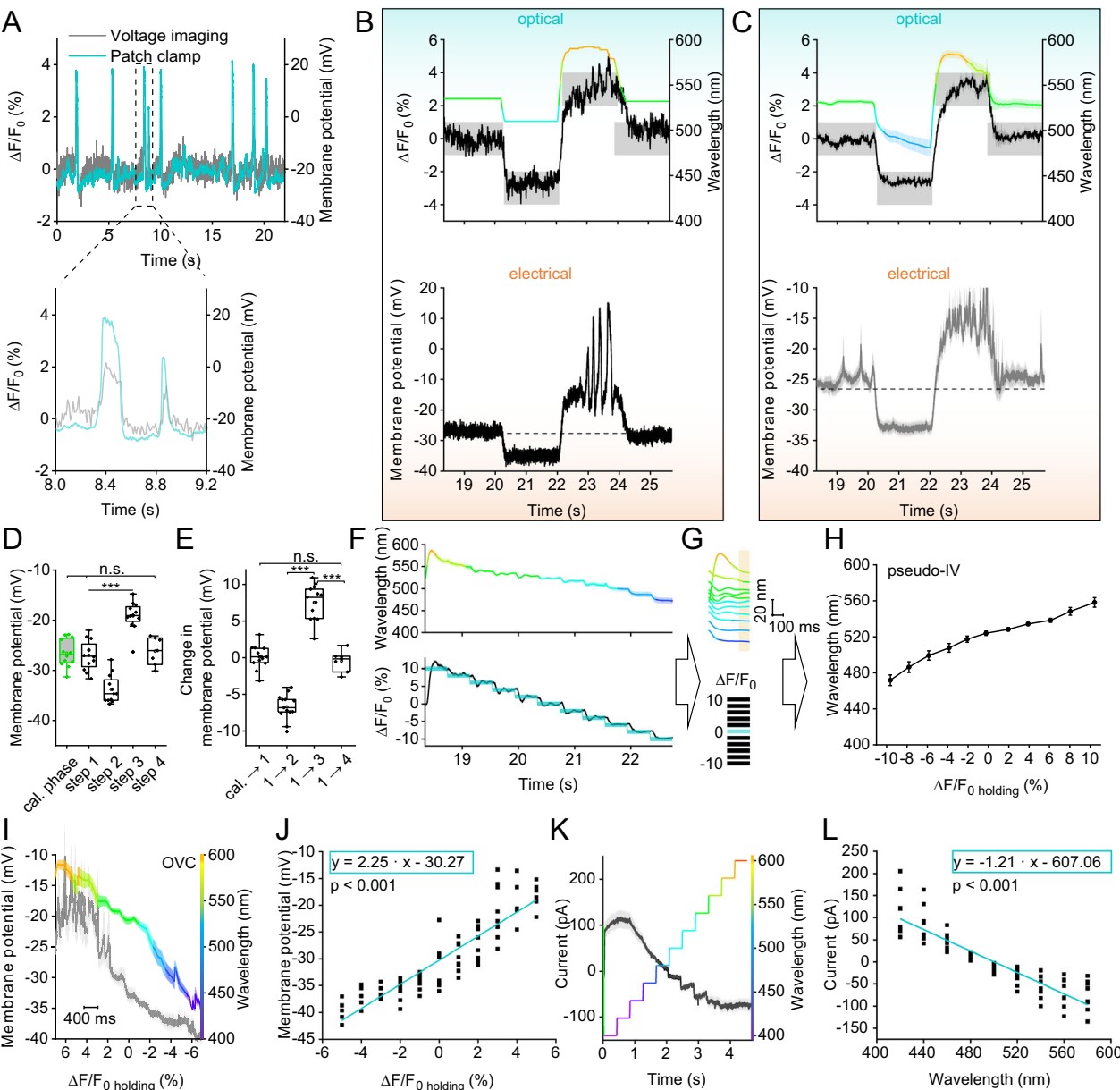

**Fig. 2 | Bi-directional optical clamping and calibration of membrane voltage and currents in BWMs. A** APs in simultaneous patch-clamp/fluorescence recordings during OVC calibration (637 nm laser + calibration wavelength). Inset: close-up. **B** Simultaneous patch-clamp/fluorescence recording, OVC four-step protocol (0, −3, 3, 0% $\Delta F/F_0$). Upper: Fluorescence recording, wavelength adaptation and tolerance ranges indicated. Lower: Corresponding patch-clamp voltage recording. **C** Overlay of mean (± S.E.M.) wavelength, fluorescence (upper) and voltage traces (lower; $n = 12$) as in **B**. **D** Statistics of data in **C**; absolute voltages of transitions during simultaneous OVC measurements ($n = 12, 12, 12, 12, 7$ for cal., steps 1, 2, 3, 4; $p = 1, 1, 1, 4.1E\text{-}6, 1.6E\text{-}7, 4.5E\text{-}16$ for cal/step 1, cal/step 4, step 1/step 4, step 1/step 2, step 1/step 3, step 2/step 3). **E** Voltage modulation for respective steps of OVC protocol in **D** ($n = 12, 12, 12, 7$ for cal→step 1, step 1→step 2, step 1→step 3, step1→step 4; and p = 7.51E-11, 1.04E-21, 1.13E-10, 1 for cal→step1/step 1→step 2, step 1→step2/ step1→step3, step1→step3/step1→step 4, cal→step1/ step 1→step 4). **F** Distinct $\Delta F/F_0$ steps, presented by the OVC (lower: blue shades, tolerance ranges; black trace, actual $\Delta F/F_0$ mean ± S.E.M.), and the required wavelengths (upper: mean ± S.E.M. nm; $n = 19$). **G, H** Determining wavelengths (≙ currents) at the end (orange shade) of each $\Delta F/F_0$ clamp step (≙ voltage) to deduce pseudo-I/V relation ($n = 19$). **I, J** Simultaneous OVC/electrophysiology experiment, setting $\Delta F/F_0$ clamp-steps, measuring resulting membrane potentials, to determine calibration regression (membrane potential/$\Delta F/F_0$; 5, 4, −3%: $n = 7$; 3, 2, 1, 0, −1%: $n = 9$; −2%: $n = 8$; −4%: $n = 6$; −5%: $n = 5$; mean ± S.E.M. (**I**)). **K, L** Voltage clamp (−24 mV) measurement of currents resulting from wavelength steps, to determine calibration regression (current/wavelength, $n = 8$, mean ± S.E.M. (**K**)). White heteroscedasticity test in (**J**, **L**, $p < 0.0001$). One-way ANOVA, Bonferroni correction (**D**, **E**), ***$p \leq 0.001$, box plots (median, 25th−75th quartiles); open dot: mean; whiskers: 1.5× IQR. $n$ refers to biological replicates (**C**−**L**). Source data are provided as a Source data file.

in response to a lack of presynaptic input, ion channels shaping muscle APs, i.e., voltage-gated $Ca^{2+}$ (EGL-19) and $K^+$-channels (SLO-2, SHK-1)[42,43], may undergo homeostatic changes to enable the observed higher muscle excitability.

*egl-19* g.o.f. mutants were shown to mediate larger currents and slowed inactivation[20,44]. We analyzed APs in the *n2368* allele using

voltage imaging (Fig. 3I, J), which revealed larger APs of longer duration than in wild type. We compared the two genotypes by generating an optical I/V-curve. In the ± 5% range (i.e., ca. −40 to −18 mV; Fig. 2I, J), *egl-19* mutants and wild type animals were similar below 0% $\Delta F/F_0$ (~resting potential) but diverged significantly once positive $\Delta F/F_0$ clamp values were reached (i.e., −28 *vs.* −20 mV;

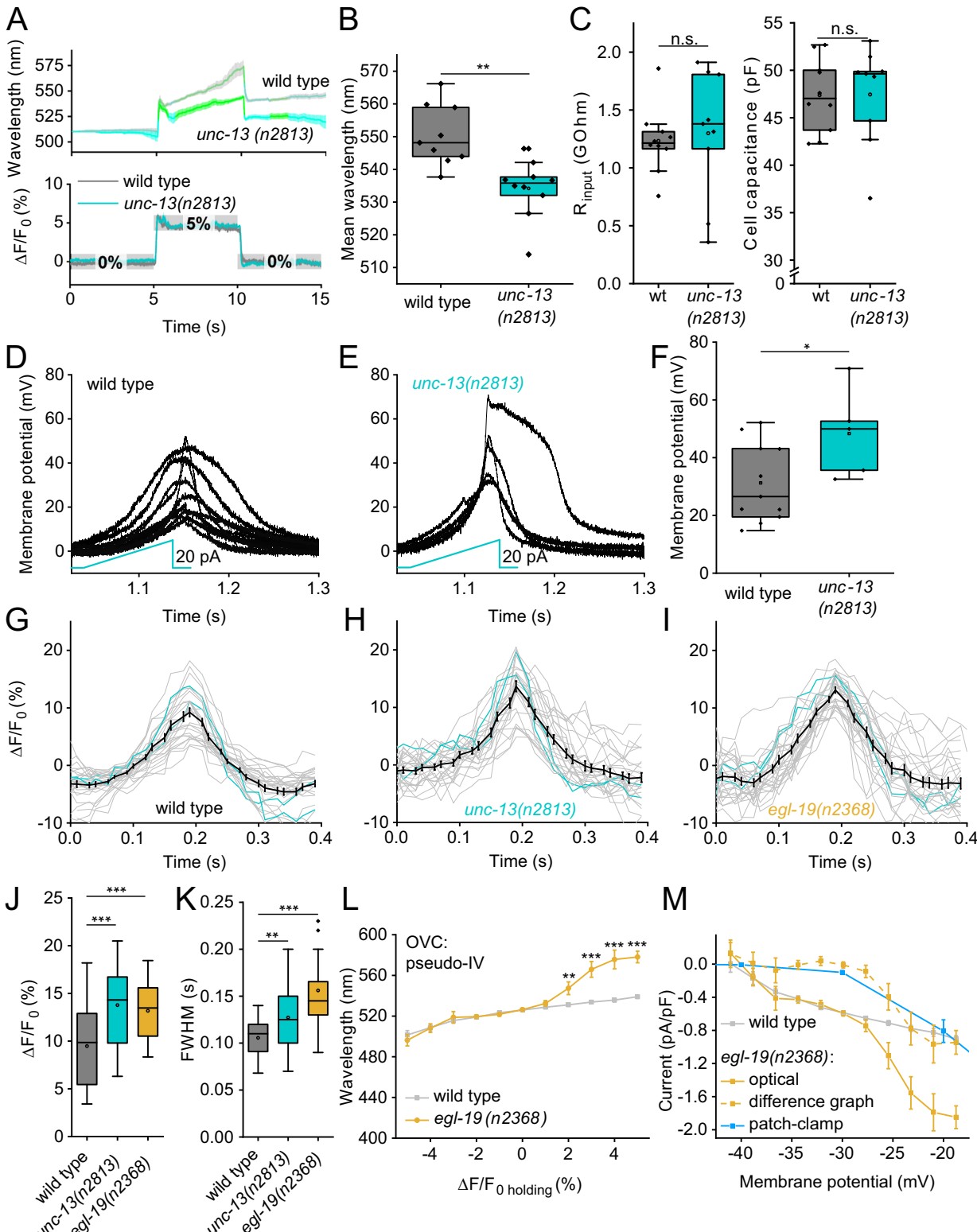

Fig. 3L and Supplementary Fig. 8E). Since we cannot block K⁺

Fig. 3L and Supplementary Fig. 8E). Since we cannot block $K^+$ channels in intact worms, we calculated the difference of wild type and mutant data to extract the additional *n2368*-mediated $Ca^{2+}$ currents (Fig. 3M). Our optically derived data, corrected for mean membrane capacitance, compared well to electrophysiological data[44] (again, difference of wild type and *egl-19* mutants; Fig. 3M). Thus, EGL-19 g.o.f. channels open at less depolarized membrane potential, compared to wild type EGL-19.

**Optogenetic current-clamp and live OVC**

Our approach enables to achieve an "optogenetic current-clamp". We wrote according software that allows to present continuous or pulsed wavelength ramps, or a single light pulse of the selected wavelength, while $\Delta F/F_0$ is recorded live (Supplementary Fig. 9A and Supplementary Code 1). This way, similar to earlier unidirectional approaches[11,12], we could induce and record APs (or inhibitory potentials) in BWMs in all-optical experiments (Supplementary Fig. 9B, C). To further extend

**Fig. 3 | Assessing altered cell physiology in mutants affecting synaptic transmission and ion channels, using the OVC. A** *unc-13(n2813)* mutant and wild type animals were subjected to OVC depolarization, to +5% $\Delta F/F_0$. Mean wavelength traces (± S.E.M.), wild type: $n = 9$, *unc-13(n2813)*: $n = 10$. **B** Mean wavelength required during the depolarization step to hold +5% $\Delta F/F_0$ ($p = 0.0012$, n−numbers as in **A**). **C** Membrane resistance was analyzed in wild type and *unc-13* mutant muscles. wild type: $n = 10$, *unc-13(n2813)*: $n = 9$. **D, E** Current ramps (0-20 pA over 100 ms; cyan) were injected into wild type and *unc-13* mutant muscles and the induced voltage increases were aligned to the peak ($p = 0.74$). wild type: $n = 11$, *unc-13(n2813)*: $n = 5$. **F** Group data of induced voltage increases in **D, E** ($p = 0.04$, n−numbers as in **D, E**). **G–I** Optical recordings of spontaneous APs in wild type animals, *unc-13(n2813)* and *egl-19(n2368)* mutants. Overlay of single (each two are highlighted in cyan) and mean (± S.E.M.) traces. wild type: $n = 6$, 27 APs analyzed; *unc-13(n2813)*: $n = 5$, 26 APs analyzed, *egl-19(n2368)*: $n = 7$, 28 APs analyzed. **J, K** Statistical analysis of peak AP amplitude (wt/*unc-13*: $p = 8.27\text{E-}4$, wt/*egl-19* $p = 3.04\text{E-}4$) and AP duration (at FWHM; wt/*unc-13*: $p = 0.0082$, wt/*egl-19* $p = 1.53\text{E-}6$) of data in **G–I**. $n$−numbers as in **G–I**. **L** All-optical $I/V$-relationship (mean ± S.E.M.) obtained in wild type ($n = 17$) and *egl-19(n2368)* mutants ($n = 14$) (2% $p = 0.018$, 3% $p = 9.36\text{E-}9$, 4% $p = 7.26\text{E-}10$, 5% $p = 7.92\text{E-}9$). **M** Data in **L** were transformed to $I/V$-relations (mean ± S.E.M.), using calibrations in Fig. 2J, L. The currents were normalized using capacitance measured in patch-clamped muscle cells (analogous to Fig. 1I). Data obtained for wild type (gray, $n = 17$) was deduced from *egl-19* data (yellow, $n = 14$), generating a difference curve (dashed, yellow), and compared to difference data obtained by electrophysiology[44] ($n = 16$) in wild type and *egl-19(n2368)* mutants (blue). Two-sided $t$ test with Bonferroni correction in (**B, C, F, J, K**). ***$p \leq 0.001$, **$p \leq 0.01$, *$p \leq 0.05$ and box plots (median, 25th–75th quartiles); open dot: mean; whiskers: 1.5× IQR. $n$−numbers refer to independent biological replicates (animals) (**A–M**). Source data are provided as a Source data file.

the applicability of the OVC, we wanted to enable the researcher to respond to observations and to dynamically adapt clamping parameters. We thus wrote software "on-the-run" allowing to select holding $\Delta F/F_0$ values during a running acquisition (Supplementary Fig. 9D and Supplementary Code 1). The software provides a live status update, whether the OVC system is on hold, adapting or if it has reached its limits (Supplementary Fig. 9E, F and Supplementary Movie 2). Using this tool, we could show that the OVC remained responsive to frequently changing live selected clamping values up to several minutes (Supplementary Fig. 9G, H).

## Establishing the OVC in cholinergic and GABAergic motor neurons

We tested if the OVC works also in *C. elegans* neurons. Cholinergic and GABAergic motor neurons are small (ca. 2–3 μm cell body, BWMs ca. 50 μm) and exhibit low absolute fluorescence. Both mCerulean (BiPOLES) and QuasAr2 fluorescence were observed in respective ganglia, including the anterior nerve ring and ventral nerve cord (Fig. 4A, B). Calibration parameters could be adopted from muscle experiments (Fig. 4C). The OVC could readily clamp neuronal voltage-dependent fluorescence between −5 and 5% $\Delta F/F_0$ (Fig. 4C–G), and also adaptive experiments were possible (Supplementary Fig. 9I, J).

## Establishing the OVC in mammalian neurons

Rodent neurons are larger, more hyperpolarized cells and display faster AP kinetics than *C. elegans* muscles and neurons. Since such cells are relevant to studies of human-like neurophysiology, we explored the utility of the OVC in rat hippocampal pyramidal neurons (organotypic slice culture; Fig. 5A), expressing QuasAr2 and soma-targeted (som)BiPOLES. After bleaching calibration (Fig. 5B), optical clamping could be achieved between ±3% $\Delta F/F_0$ (Fig. 5B, C) with the OVC protocol as used in *C. elegans*. Importantly, in the absence of somBiPOLES, monochromator light did not lead to modulation of QuasAr2 fluorescence (Fig. 5D), while electrically evoked potential shifts (100 mV depolarization step, from −74.5 mV to +25.5 mV holding voltage) caused clear increases of QuasAr2 fluorescence (ca. 21% $\Delta F/F_0$; Fig. 5E). In cells expressing QuasAr2 and somBiPOLES, the ±3% $\Delta F/F_0$ OVC protocol caused hyperpolarizing and depolarizing potential jumps (ca. −4 mV and +3 mV, respectively; Fig. 5F, G). These were small, compared to earlier experiments using somBiPOLES, where 595 nm light application caused depolarization of up to 30 mV, while 400 nm light clamped cells to the Cl⁻ reversal potential[27]. Possibly, due to the more negative resting potential in mammalian neurons compared to *C. elegans* cells, 637 nm activation of Chrimson causes stronger effects, and thus the OVC triggers more compensatory GtACR2 currents. When we expressed somBiPOLES only, 637 nm laser light depolarized hippocampal neurons (via Chrimson) by ca. 21.5 mV, which could be partially counteracted by GtACR2 activation using 530 nm compensation light (Fig. 5H). However, due to the high conductance of GtACR2, shunting effects likely prevented further optical hyper- or depolarization. Thus,

in mammalian neurons, the OVC works with a limited range, likely due to different voltage and ion conditions in the resting state, and probably due to higher relative expression levels, as compared to *C. elegans*.

## Dynamic suppression of APs in pharyngeal muscle

Thus far, we imposed fixed voltage steps to cells. To see if the OVC can dynamically counteract spontaneous activity, as opposed to shunting hyperpolarization, we turned to a periodically active muscular pump, the *C. elegans* pharynx (Fig. 6A).

This feeding organ exhibits periodic, ca. 4 Hz APs[20]. Closing of the grinder, a structure used to crush bacteria, was forced by activation of *Gt*ACR2 (400 nm), and could be observed by QuasAr2 fluorescence (Supplementary Fig. 10A). Illumination with only the 637 nm laser caused grinder opening. The OVC could clamp the pharynx statically between −5 and 5% $\Delta F/F_0$ (Supplementary Fig. 10B–D), and dynamically follow its activity to keep it at 0% $\Delta F/F_0$ (Fig. 6B, C and Supplementary Movie 3). Optically observed APs showed a rise time constant of around 15 ms and duration of ca. 150 ms (at full-width-half-maximum (FWHM))[20]. To clamp these APs dynamically, i.e., to counteract de- and repolarization phases quickly enough, we set the OVC software parameters to respond with more pronounced wavelength changes to a given fluorescence change (increased integral gain; "Methods," Eq. 6), and reduced the tolerance range to ± 0.005%. During calibration, APs showed 26.0 ± 1.6% $\Delta F/F_0$ mean amplitude and 118.7 ± 7.3 ms duration at FWHM (Fig. 6B, C, E, F). Upon clamping, all subsequent attempts to spike were dynamically counteracted and almost completely suppressed (Fig. 6B–F). Clamping was rapid: APs occurring at clamping onset were shortened by 65%, while the OVC traversed the full wavelength spectrum within 20 ms (Fig. 6D). Signal amplitude was significantly reduced to ca. 5 ± 0.7% (Fig. 6E, F), as was its duration (to 83.14 ± 8.8 ms at FWHM). The greatly reduced voltage signals also resulted in the suppression of pump events (Fig. 6G). This finding supports the effectiveness of OVC-mediated AP suppression. As shown above in BWMs, APs, associated with the opening of the grinder, could be elicited using the current clamp mode (Supplementary Fig. 10E, F).

## Dynamic clamping of APs in the GABAergic motor neuron DVB

*C. elegans* exhibits a rhythmic motor program to move and expel gut contents[29–32]. Expulsion muscle contraction is regulated by an intestinal pacemaker and by GABAergic motor neurons DVB and AVL (Fig. 7A). Ca²⁺ imaging of DVB[45] revealed activity reminiscent of APs, thus far observed in only few *C. elegans* neurons[46]. DVB voltage imaging showed APs (7.7 ± 1% $\Delta F/F_0$, 500 ± 60 ms at FWHM) at regular time intervals of 40-50 s, that were followed by posterior body contraction (Fig. 7B). Using parameters as for the pharynx, DVB APs were clamped to significantly reduced amplitude and duration (Fig. 7C, D and see Supplementary Fig. 10G for single trace). DVB allowed calibrating the neuronal OVC by patch-clamp electrophysiology[47]: resting potential was −49.0 ± 8.9 mV (Fig. 7E, F), and increasing current steps (1 pA)

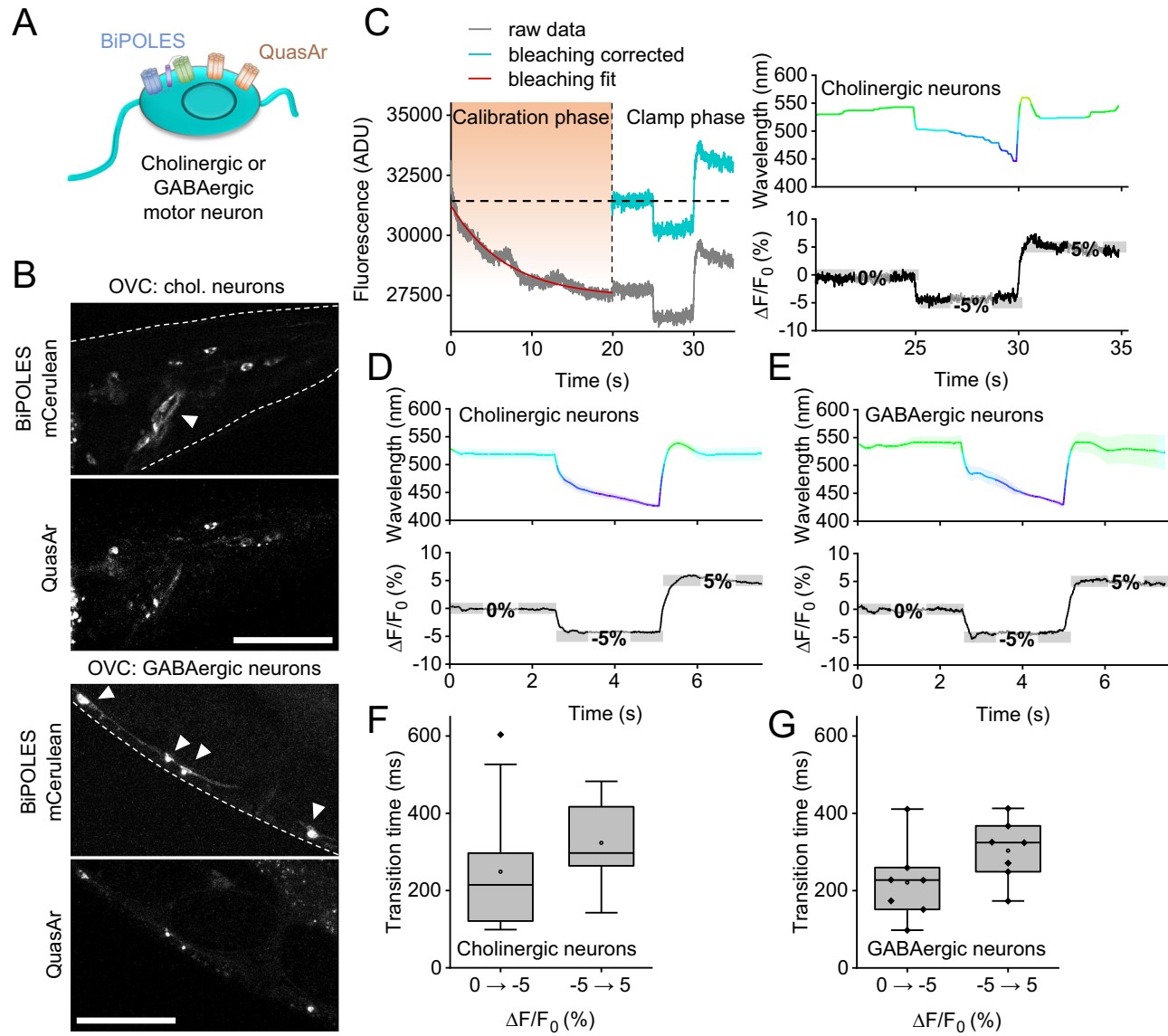

**Fig. 4 | Bi-directional clamping of voltage-dependent fluorescence in *C. elegans* neurons. A** OVC in cholinergic or GABAergic neurons. **B** Confocal fluorescence z-projections of BiPOLES (mCerulean) and QuasAr2 expression. Arrowheads: Neuronal cell bodies. Scale bars: 50 μm. Representative image (from *n* = 11 (cholinergic) or 16 (GABAergic neurons)). **C** OVC protocol (holding values 0, −5, 5% $\Delta F/F_0$) in cholinergic neurons, single recording. **D, E** Mean (±S.E.M.) traces of the OVC in cholinergic (**D**; *n* = 14), or GABAergic (**E**, *n* = 7) neurons, respectively, holding values: 0, −5, 5% $\Delta F/F_0$. **F, G** Transition times for 5 and 10% $\Delta F/F_0$ steps (**F**: cholinergic, **G**: GABAergic neurons, *n*−numbers as in **D**, **E**); box plots (median, 25th–75th quartiles), open dot: mean, whiskers: 1.5× IQR. *n*−numbers refer to independent biological replicates (animals) (**D**–**G**). Source data are provided as a Source data file.

evoked APs (−23.4 mV threshold, depolarization to 26.3 mV, 359 ms duration (FWHM), −38.5 mV after-hyperpolarization; n = 5; Fig. 7F). DVB AP amplitude thus is ca. 50 mV. Although fluorescence baseline may be altered by the 637 nm laser and Chrimson activation, this did not evoke APs. Thus, the QuasAr2 signal in DVB, i.e., 7.7% $\Delta F/F_0$ for an AP, corresponds to ca. 50 mV (calculating from threshold, or up to 75 mV, from resting potential).

## Discussion

Here, we established the first, to our knowledge, all-optical voltage-clamp approach to date. We demonstrated its performance in various excitable cell types in intact animals (*C. elegans*) and tested it in mammalian hippocampal neurons. In *C. elegans*, the OVC allowed reliable clamping of voltage in two muscular organs and three neuron types via QuasAr2 fluorescence read-out and bidirectional optogenetic actuation *via* BiPOLES. The OVC could further detect the altered postsynaptic excitability of *unc-13* mutants, a

homeostatic response to reduced presynaptic input, and allowed deducing all-optical I/V-relationships that provided insight into altered functionality of the L-type VGCC EGL-19 in a g.o.f. mutant, matching electrophysiological data. Using the OVC, we did not only control resting membrane potential, but could also dynamically clamp spontaneous rhythmic activity and APs in pharyngeal muscle and in the motor neuron DVB.

The OVC operates at sampling rates up to 100 Hz on a typical PC with moderate computing power. This orchestrates the communication between camera and monochromator, while in parallel running image acquisition and providing live computation of bleaching-corrected $\Delta F/F_0$ values, fed into a decision tree and Integral-control algorithm. An alternative algorithm with PID controller and Kalman filter showed similar performance. While the OVC at present is slower than standard patch-clamp electrophysiology, it is not necessarily less sensitive or accurate, within the range of experimental variation, and given variations imposed by dissection required for electrophysiology.

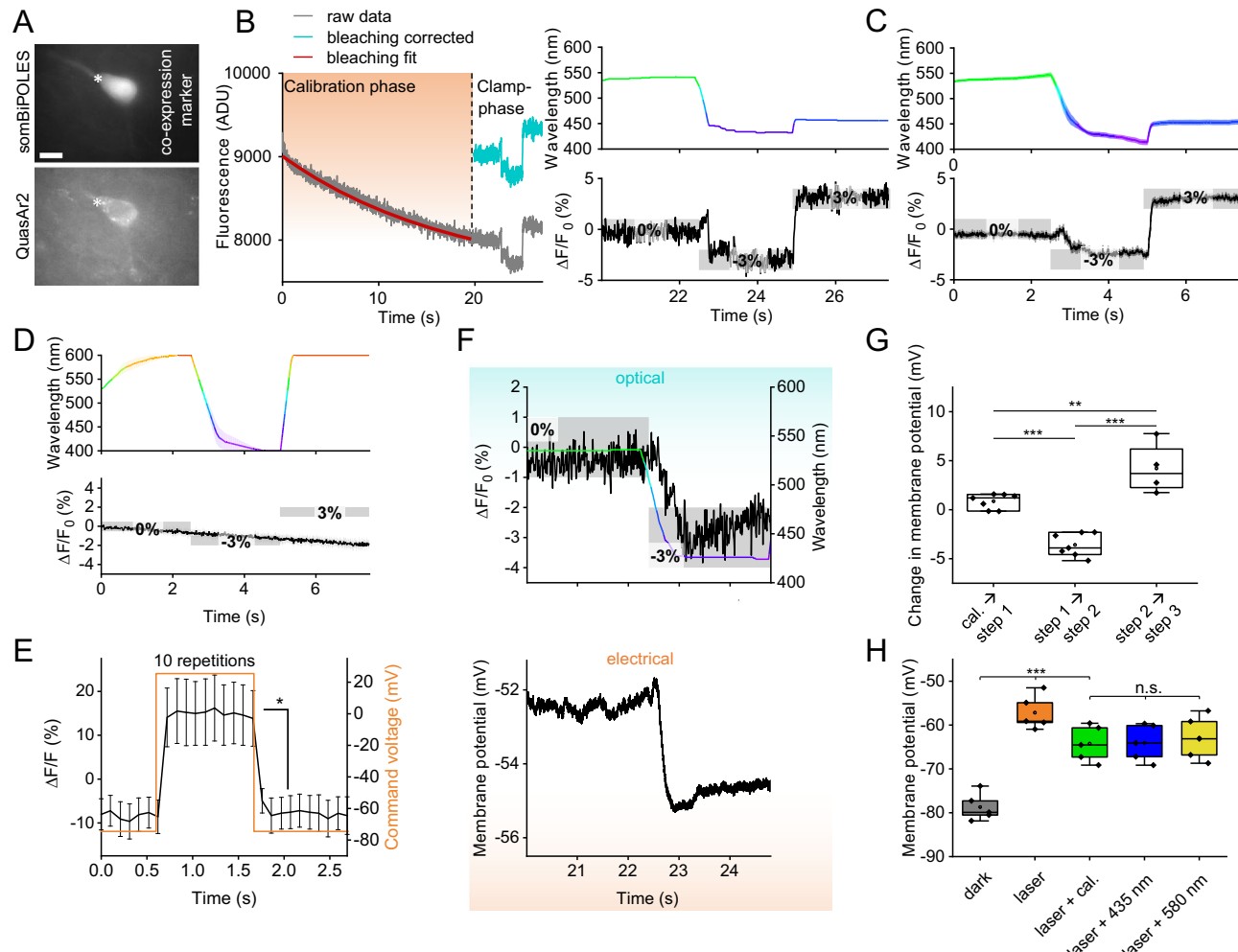

**Fig. 5 | Establishing the OVC in rat hippocampal pyramidal neurons.**
**A** Fluorescence micrographs of somBiPOLES (CFP co-expression marker) and QuasAr2 expression in a neuron (asterisk). Scale bar: 10 μm. Representative image (from $n = 3$). **B** OVC three-step protocol (0, −3 and 3% $\Delta F/F_0$). Right: Wavelength shown in the respective color (upper), holding values and tolerance range (gray boxes, lower panel) are indicated for each step. Left: Orange shade marks transition/calibration period to reach tolerance range. **C** Overlay of mean (±S.E.M.) wavelength (upper panel) and fluorescence traces (lower panel; $n = 12$; holding values: 0, −3, 3% $\Delta F/F_0$). **D** Average fluorescence data (lower panel) for cells expressing QuasAr2 only, while the OVC attempts to run a 0, −3, 3% $\Delta F/F_0$ protocol ($n = 5$; upper panel: Monochromator wavelength). **E** QuasAr2 fluorescence during electrically evoked 100 mV depolarization step, from −74.5 mV to +25.5 mV holding voltage ($n = 3$, 10 steps analyzed per cell, $p = 0.046$). **F** Simultaneous patch-clamp

(voltage, lower panel) and fluorescence recording (upper panel) with indicated wavelength adaptation and tolerance ranges. **G** Statistical analysis of voltage modulation between transition events during simultaneous patch clamp/OVC measurements (step 1 → step 2/cal. → step 1: p = 1.62E-4; step 1 → step 3/cal. → step 1: $p = 0.00578$; step 1 → step 3/step 1 → step 2: $p = 2.96E-6$; cal. → step 1: $n = 7$, step 1 → step 2: $n = 7$, step 1 → step 3: $n = 4$). **H** Modulation of membrane voltage in cells expressing somBiPOLES only, in response to different light application, as indicated (laser/dark: $p = 7.07E-7$; laser + cal./dark: $p = 1.97E-4$; $n = 5$). Two-sided $t$ test with Bonferroni correction in **E**. One-way ANOVA with Bonferroni correction in **G**, **H**. ***$p \le 0.001$, **$p \le 0.01$, *$p \le 0.05$. In **G**, **H**, box plots (median, 25th−75th quartiles); open dot: mean; whiskers: 1.5× IQR. $n$−numbers refer to independent biological replicates (animals) (**C**−**E**, **G**, **H**). Source data are provided as a Source data file.

The OVC outperforms electrophysiology in terms of non-invasiveness, throughput and ease of application.

Though activation of Chrimson by QuasAr2 excitation light evoked currents, this effect could be counterbalanced in *C. elegans* by the bidirectionality of BiPOLES, using compensating GtACR2 activation. Despite this low level opening of BiPOLES channels, no significant effects on membrane resistance or membrane potential were observed, intrinsic neuronal activity and cellular excitability were normal, muscles fired regular APs, locomotion behavior was unaltered, and also spontaneous activity in pharyngeal muscle or the DVB neuron were unaffected.

The voltage range covered by the OVC differed between muscles and neurons: 10% $\Delta F/F_0$ change corresponded to about 22 mV in BWMs, and to about 65 mV in DVB. Differences in membrane potential and AP amplitude depend on the different ion channels and gradients

present in the two cell types, and input resistance. The different range of $\Delta F/F_0$ fluorescence of QuasAr2 in muscles and neurons may also correspond to different levels of QuasAr2 protein in plasma membrane vs. intracellular membranes of the two cell types. Fluorescence from the latter would not contribute to voltage-dependent $\Delta F/F_0$, while it would increase overall fluorescence at rest.

BiPOLES mediated currents in a range of approx. 190 pA, comparable to common optogenetic tools[33,34,48], but falling behind the effects of individually expressed ACR2[19]. In terms of accuracy and in line with previous patch-clamp measurements in BWMs, the examined (patched) cells show a normal distribution in resting potential (ca. −23 to −27 mV), which remains unchanged during the calibration phase. Since our system is based on relative changes in fluorescence, which originate from slightly varying resting potentials, set clamp values may be subject to small variations. The OVC is

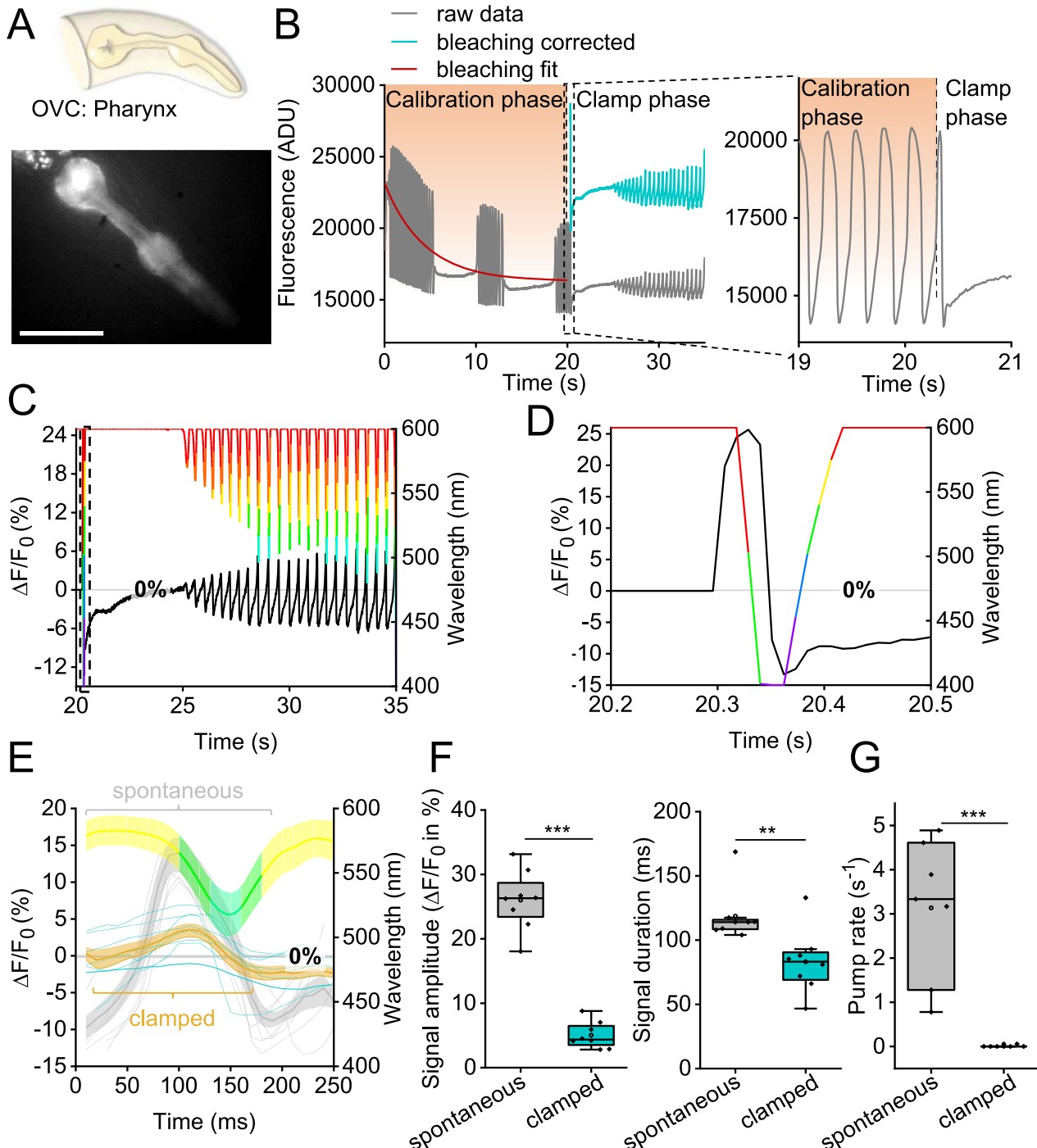

**Fig. 6 | All-optical clamping of APs in pharyngeal muscle. A** The pharynx, a muscular pump used for feeding, expressing BiPOLES and QuasAr2 (fluorescence, representative image (from $n = 8$)). Scale bar is 50 µm. **B–D** OVC experiment in pharyngeal muscle, holding fluorescence at 0% $\Delta F/F_0$ and suppressing APs. **B** Original trace, calibration and clamping. Inset: close-up of boxed region, transition calibration to clamp phase. ADU: analog-to-digital units. **C** Overlay: wavelength and $\Delta F/F_0$ traces during clamp phase. **D** Close-up of box in **C**: OVC counteracting first AP during clamp phase. **E** Aligned traces of spontaneous (light gray, $n = 8$ animals, 5 APs each) and clamped (blue) pharyngeal fluorescence signals. Mean

wavelength chosen by the system shown in respective color. **F** Statistics of data in **E** and additional APs ($n = 8$ animals, 10 APs each; amplitude $p = 1.45\text{E-}8$, duration $p = 0.0078$), fluorescence voltage signal amplitude and duration at FWHM. **G** Pumping, observed visually, was suppressed by dynamic OVC clamping ($n = 7$; $p = 2.028\text{E-}4$). In **F**, **G**: box plots (median, 25th–75th quartiles), open dot: mean, whiskers: 1.5× IQR. In **F**, **G**, two-sided $t$ test with Bonferroni correction. ***$p \leq 0.001$, **$p \leq 0.01$. $n$ refers to biological replicates (**E–G**). Source data are provided as a Source data file.

not a one-to-one replacement for electrophysiology: Membrane potential is mapped as a relative change using fluorescence, thus requiring calibration measurements to deduce absolute values. The rhodopsins used are not as persistent as a patch-clamp electrode

and limited by the cells' reversal potentials for the respective ions. However, in the range of physiological voltage fluctuations in the respective cell type, the OVC is robust and enables generating data with the advantages of a purely optical, contact-less system. This

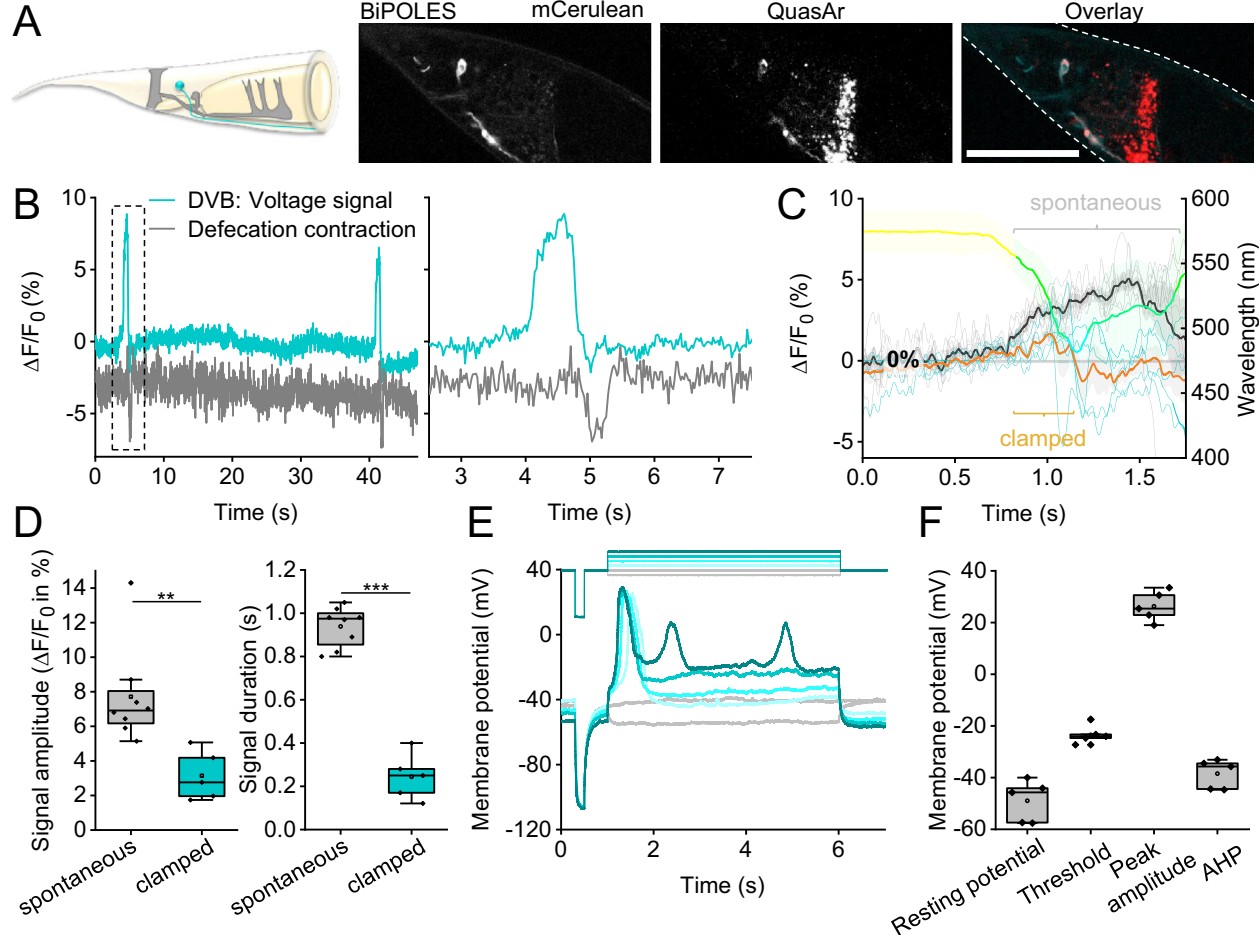

**Fig. 7 | All-optical clamping of APs in the enteric motor neuron DVB.**
**A** GABAergic motor neuron DVB (cyan), innervating enteric muscles (gray, upper panel). Below: confocal *z*-projection, BiPOLES and QuasAr2 in DVB, and overlay; scale bar: 50 µm. **B** Voltage fluorescence signals, spontaneous APs in DVB (blue trace). Expulsion muscle contraction, deduced from DVB movement (gray trace). Right panel: Close-up of boxed region. **C** Overlay of spontaneous (light gray traces, *n* = 8; mean: black) and clamped (blue traces, *n* = 5; mean: orange) DVB voltage signals. Mean wavelength set by OVC shown in respective color. **D** Statistics of data in **C**, mean voltage fluorescence signal amplitude and total duration (amplitude

*p* = 0.0073, duration *p* = 7.97E-8). **E** Current clamp recordings (*n* = 5, lower panel) of DVB in dissected animals, following indicated (upper panel) hyperpolarizing (gray) and depolarizing (cyan) current injection (1 pA steps). **F** Group data of **E**: resting potential, threshold, peak amplitude, duration at FWHM, and after-hyperpolarization (AHP). In **D**, **F**: box plots (median, 25th–75th quartiles), open dot: mean, whiskers: 1.5× IQR. In **D**, two-sided *t* test with Bonferroni correction. ***\*p* ≤ 0.001, \*\*p* ≤ 0.01. *n* refers to biological replicates (**C**–**F**). Source data are provided as a Source data file.

enables acquiring I/V-relationships, and to some extent (since without feedback), current clamp.

In mammalian neurons, optical voltage control was currently limited to a few mV. This was likely due to the resting potential being close to the Cl⁻-reversal potential, but possibly also due to BiPOLES expression levels, as imaging light affected membrane conductance. Thus, in comparison to *C. elegans*, more activation of GtACR2 is required to counteract Chrimson-based depolarization, causing membrane leak. Minimizing the optical crosstalk of voltage sensor and BiPOLES is especially important in mammalian neurons. Further red-shifted voltage sensors and blue-shifted BiPOLES variants are thus required for future OVC experiments. Light-driven ion pumps, provided they become as powerful as the channels used, could be a future alternative.

The speed of the OVC, characterized by transition times of about 90–150 ms for the relevant induced Δ*F*/*F*₀ steps, is currently limited by the software that operates at a maximum sampling rate of 100 Hz. The software currently runs within µManager on a PC. Running the software on an integrated circuit, like a field programmable gate array, and using small ROIs, may enable faster sampling rates up to 1 kHz. The goal would be to identify the optimal sampling rate, thus bringing the

transition time closer to the open loop system time constant of approx. 20 ms (50 Hz system frequency) and optimize control quality. In general, sampling rates should exceed the system frequency by at least a factor greater than 2, but ideally by 6–20 (ca. 300–1000 Hz)[49,50]. The optimal sampling rate is not exclusively dependent on that system frequency, but is also limited by the necessary computing time of the processor and the photon count at shorter exposure times. Considering the high photon count at 10 ms exposure, and assuming a linear relation of exposure time and photon count, we expect that the latter will still be sufficient at theoretical sampling rates of up to 1 kHz (assuming 185,000 captured photons, accuracy <0.4%; Supplementary Table 1). In addition, alternative, or even future optogenetic tools with faster kinetics might overcome the ultimate speed limitation imposed by the OVC's current tool combination and inherent system frequency. Here a 10x faster variant of Chrimson[25,26], and the 20x faster ZipACR were described[49]. That rhodopsin channels show some inactivation during prolonged illumination is not a concern, because the OVC feedback loop can counteract progressive inactivation of BiPOLES components, at least until one of those would desensitize completely. We observed no problems in measurements lasting up to three minutes.

The OVC detected altered muscular excitability in *C. elegans unc-13* mutants, likely caused by changes in ion channel physiology, and it could directly confirm altered channel excitability and currents through a g.o.f. variant of the EGL-19 VGCC. This shows that the OVC represents an approach to enable all-optical, contact-less high-throughput screening applications, e.g., of compound libraries targeted at ion channels. Testing its use in mammalian cell types other than neurons, where such ion channels can be expressed individually, and where the resting membrane potential is less depending on the Cl⁻ gradient, will facilitate this approach.

The establishment of the OVC paves the way for all-optical control of individual neurons in freely behaving transparent animals like *C. elegans*. Accuracy of fluorescence quantification (and thus OVC feedback) requires the cell to remain in focus, which can be achieved with the OVC, while it is impossible to keep a patch-clamp electrode physically attached to such a small animal. Using the OVC on neurons controlling behavior will allow fine-tuning of behavioral aspects, and enable understanding how activity of the individual cell regulates them. Online behavioral analysis (e.g., extent of body bending, locomotion velocity), may be used for feedback to the neuron such that behavior can be dynamically controlled.

Transferring the OVC to other model organisms may require modifications depending on the respective cellular properties. Extending electrophysiology applications, and provided the monochromator light is projected via a digital micromirror device, the OVC should enable efficient space clamp, as well as dynamic local clamping in neuronal processes. The OVC software is universally applicable, as it can be adapted to other GEVI-optogenetic actuator combinations. The OVC broadens applicability of optogenetics as it allows modulation in closed-loop, to better adapt to the variable activity patterns found in living organisms. Dynamic responsiveness is also advantageous with regard to future therapeutic applications, for example in acute control of seizures[51], or in adaptive deep brain stimulation, as it would allow adjusting the therapy to the need of the patient[52].

## Methods

### Transgenic *C. elegans* strains

*C. elegans* were cultivated at room temperature (21 °C) on nematode growth medium (NGM) plates, seeded with *E. coli* OP50-1 strain[41]. The following strains were used or generated, and are available upon request: **ZX2476**: *zxEx1139[pmyo-3::QuasAr2; pmyo-2::CFP]*, **ZX2482**: *zxEx1145[pmyo-3::QuasAr2; pmyo-2::CFP]; zxIs5[punc-17::ChR2(H134R)::yfp;lin-15 +]*, **ZX2483**: *zxEx1146[punc-17::ACR2::eYFP; pmyo-3::QuasAr2; pelt-2::GFP]*, **ZX2586**: *zxEx1228[punc-17::GtACR2::mCerulean::βHK::Chrimson; pelt-2::GFP]*, **ZX2714**: *zxEx1250[punc-17::GtACR2::mCerulean::βHK::Chrimson; pmyo-3::QuasAr2; pelt-2::GFP]*, **ZX2753**: *zxEx1266[pmyo-3::GtACR2::mCerulean::βHK::Chrimson; pmyo-3::QuasAr2; pmyo-2::CFP]*, **ZX2755**: *zxEx1268[punc-47::QuasAr2::GFP; pmyo-2::CFP]*, **ZX2826**: *zxEx1282[pmyo-2::QuasAr2; pmyo-2::GtACR2::mCerulean::βHK::Chrimson; pmyo-3::CFP]*, **ZX2827**: *zxEx1283[punc-17::GtACR2::mCerulean::βHK::Chrimson; punc-17::QuasAr2; pelt-2::GFP]*, **ZX2828**: *zxEx1284[punc-47::QuasAr2::GFP; punc-47::GtACR2::mCerulean::βHK::Chrimson; pmyo-2::CFP]*, **ZX2876**: *zxIs139[pmyo-3::GtACR2::mCerulean::βHK::Chrimson; pmyo-3::QuasAr2; pmyo-2::CFP]*, **ZX2935**: *unc-13(n2813); zxIs139[pmyo-3::GtACR2::mCerulean::βHK::Chrimson; pmyo-3::QuasAr2; pmyo-2::CFP]*; **ZX3074**: *egl-19(2368); zxIs139[pmyo-3::GtACR2::mCerulean::bHK::Chrimson; pmyo-3::QuasAr; pmyo-2::CFP]*. These strains are available upon request from A. Gottschalk.

### Molecular biology

Plasmids pAB4 (p*unc-17*::ACR2::eYFP), pAB16 (p*myo-3*::QuasAr; Addgene plasmid #130272), pAB17 (p*unc-17*::QuasAr), pAB23 (p*tdc-1s*::QuasAr::GFP) and pNH12 (p*myo-2*::MacQ::mCitrine) were described earlier[19,20]. **pAB26 (p*unc-17*::GtACR2::mCerulean::βHK::Chrimson)** was generated by Gibson Assembly based on RM#348p (p*unc-17*; a gift from J. Rand) and pAAV-hSyn-BiPOLES-mCerulean (Addgene plasmid #154944), using *Nhe*I and primers 5′-attttcaggaggaccctttggATGGCATCACAGGTCGTC-3′ and 5′-ataccatggtaccgtcgacgTCACACTGTGTCCTCGTC-3′. **pAB27 (p*myo-3*::GtACR2::mCerulean::βHK::Chrimson)** was generated *via* Gibson Assembly based on pDD96.52 (p*myo-3*, Addgene plasmid #1608) and pAAV-hSyn-BiPOLES-mCerulean, using *Bam*HI and primers 5′-actagatccatctagagATGGCATCACAGGTCGTC-3′ and 5′-ttggccaatcccgggCACTGTGTCCTCGTCCTC-3′. **pAB28 (p*unc-47*::QuasAr::GFP)** was generated by Gibson Assembly based on pMSM08 (p*unc-47*::eGFP::MmBoNTB) and pAB23 (p*tdc-1s*::QuasAr::GFP), using *Xma*I, *Msc*I and primers 5′-ttacagcaccggtggattggATGGTAAGTATCGCTCTG-3′ and 5′-ttctacgaatgctcctaggcCTATTTGTATAGTTCATCCATGC-3′. **pAB29 (p*myo-2*::QuasAr)** was generated by Gibson Assembly based on pNH12 (p*myo-2*::MacQ::mCitrine) and pAB16 (p*myo-3*::QuasAr), using primers 5′-caccgagtgaGAAGAGCAGGATCACCAG-3′, 5′-tgcagagcgatacttaccatCCCCGAGGGTTAAAATGAAAAG-3′, 5′-ATGGTAAGTATCGCTCTGCAG-3′ and 5′-cctgctcttctcaCTCGGTGTCGCCCAGAATAG-3′. **pAB30 (p*myo-2*::GtACR2::mCerulean::bHK::Chrimson)** was generated by Gibson Assembly based on pNH12 (p*myo-2*::MacQ::mCitrine) and pAAV-hSyn-BiPOLES-mCerulean, using *Bam*HI, *Hind*III and primers 5′-ggacgaggacacagtgtgaaAAGAGCAGGATCACCAGC-3′ and 5′-agacgacctgtgatgccatgCCCCGAGGGTTAAAATGAAAAG-3′. **pAB31 (p*unc-47*::GtACR2::mCerulean::bHK::Chrimson)** was made by Gibson Assembly based on pAB28 (p*unc-47*::QuasAr::GFP) and pAAV-hSyn-BiPOLES-mCerulean, using *Age*I, *Eco*RI and primers 5′-acatttatttcattacagcaATGGCATCACAGGTCGTC-3′ and 5′-agcgaccggcgctcagttggTCACACTGTGTCCTCGTC-3′. These plasmids available upon request from A. Gottschalk.

For mammalian neuronal expression, the coding sequence of QuasAr2[53] (Addgene #107705) was cloned together with a trafficking signal (ts: KSRITSEGEYIPLDQIDINV) and an ER-export signal (ER: FCYENEV) from the Kir 2.1 channel[54,55] into an AAV2-backbone behind a human synapsin (hSyn) promoter resulting in **pAAV-QuasAr2-ts-ER**. These plasmids are available upon request from S. Wiegert.

### OVC voltage imaging experiments

For voltage imaging experiments, animals were supplemented with all-*trans* retinal (ATR; Sigma-Aldrich, USA): One day prior to experiments, transgenic L4 stage animals were transferred to NGM plates, seeded with OP50 bacterial suspension mixed with ATR (stock in ethanol). To avoid interfering fluorescence of unbound ATR, its concentration was adjusted for each tissue. Final ATR concentrations (mM): BWMs (0.01), pharynx (0.03), cholinergic neurons (0.1), GABAergic neurons (0.005). Animals were immobilized with polystyrene beads (0.1 μm diameter, at 2.5% w/v, Sigma-Aldrich) on 10% agarose pads (in M9 buffer). Voltage-dependent fluorescence of QuasAr2 was excited with a 637 nm red laser (OBIS FP 637LX, Coherent) at 1.8 W/mm$^2$ and imaged at 700 nm (700/75 ET Bandpass filter, integrated in Cy5 filter cube, AHF Analysentechnik), while optogenetic actuators (BiPOLES, *Gt*ACR2 or ChR2(H134R)) were activated using a monochromator (Polychrome V, Till Photonics/Thermo Scientific), set to emit light from 400 to 600 nm at 300 μW/mm$^2$. Imaging was performed on an inverted microscope (Zeiss Axio Observer Z1), equipped with a ×40 oil immersion objective (Zeiss EC Plan-NEOFLUAR ×40/N.A. 1.3, Oil DIC ∞/0.17), a laser beam splitter (HC BS R594 lambda/2 PV flat, AHF Analysentechnik), a galilean beam expander (BE02-05-A, Thorlabs) and an EMCCD or an sCMOS camera (Evolve 512 Delta, Photometrics, or Kinetix 22, Teledyne). All OVC experiments were performed at up to 100 fps with 10 ms exposure and a binning of 4 × 4 (computer specifications: 24 GB RAM, AMD FX-8150 Octa-core processor (3.6 GHz), NVIDIA GeForce GT 520). To induce pharyngeal pumping, animals were incubated in 3 μl serotonin (20 mM, in M9) for 3 min prior to experiments.

## OVC software

The OVC control software was written in Beanshell, the scripting language used by the open source microscopy software µManager v. 1.4.22[35]. Scripts are provided in supplementary information (Supplementary Code 1). Experiments were initiated via the µManager script panel. For support of Polychrome V related commands, a copy of the Java archive file *TILLPolychrome.jar* must be placed into the MMPlugins folder and the dynamic link library file *TILLPolychromeJ.dll* into the Sys32 folder (both provided by Till Photonics). The main OVC software is compatible with all cameras supported by µManager. Before running the software, simple rectangular ROIs (to save computation capacity, Supplementary Fig. 1B) must be selected in the live image mode by using the ROI button in the µManager main window. Once a ROI is selected, the script can be executed via the script panel GUI. An input tab prompts the user to select the OVC- and acquisition parameters. The software allows to define a holding $\Delta F/F_0$ value (in %) and the number of frames for each of the steps (three- or four-step-protocol). One can further select the number of frames for the calibration period, the tolerance range (in %; range in which the actual value may fluctuate around the target $\Delta F/F_0$, and in which limits the control variable wavelength is set to hold), the algebraic sign of the increment factor that decides whether to increase or decrease the wavelength with respect to the current $\Delta F/F_0$, the starting wavelength and the wavelength limits (in nm). The free choice of these parameters ensures that the program can also be used for other and/or future combinations of optogenetic tools with different spectral properties. For acquisition, light intensity of the Polychrome V (in %), exposure time of the camera (in ms) and binning can be selected.

During calibration, acquired gray values are stored in an array and used to evaluate the parameters for the exponential decay function to correct for bleaching of the voltage sensor, live during the clamping phase (Eqs. 1–3; based on the ImageJ Plugin "Correctbleach"[56,57]). At the last time point of the calibration phase, the first 50 bleaching corrected gray values of the recording are used to calculate $F_0$. Subsequently, bleaching corrected $\Delta F/F_0$ values are calculated at each time point of the clamping phase (Eq. 4). Once the system has access to the bleaching-corrected $\Delta F/F_0$ values, it feeds them into a decision tree algorithm, where they are compared to a desired holding $\Delta F/F_0$ value. Depending on the sign of the difference, it decides, whether to shift the wavelength blue or red (mirrored by the sign of the increment i[k]). Wavelength adaptation occurs *via* an implemented I-controller (integral gain $K_I$ was empirically chosen to fit the desired system-behavior) that maps the "I" dynamics by buffering the manipulated variable (wavelength $\lambda$) to compensate for permanent control deviation, which would otherwise occur by using a sole P-controller (Supplementary Fig. 1A). That means that the difference (error) between command and current $\Delta F/F_0$ determines the wavelength change of the monochromator – hence, the current wavelength results from the previous plus the respective wavelength change at each time point (Eqs. 5 and 6). The software comes with a three- or four-step protocol, where among others, the desired holding values (within a tolerance range) and number of frames for each step can be selected by the user (Supplementary Fig. 1C and Supplementary Code 1). The system changes the wavelength only if the tolerance range is exceeded.

$$\mathrm{Exp.\ offset} = a*e^{-b*\mathrm{frame}} + c \tag{1}$$

$$\mathrm{ratio} = \exp.\mathrm{offset}_0 \div \exp.\mathrm{offset}_\mathrm{frame} \tag{2}$$

$$F_\mathrm{correct} = F_\mathrm{frame}*\mathrm{ratio} \tag{3}$$

$$\Delta F/F_{0\,\mathrm{correct}} = ((F_\mathrm{correct} - F_0)/F_0)*100 \tag{4}$$

$$e = |\Delta F/F_{0\,\mathrm{holding}} - \Delta F/F_0| \tag{5}$$

$$\lambda = \lambda_\mathrm{frame-1} + K_I*e*i[k] \tag{6}$$

Exp. offset: exponential offset; *a*, *b*, *c*: exponential fitting parameters; Ratio: ratio between first and current exponential offset; $F_\mathrm{correct}$: Bleaching-corrected gray values; $F_\mathrm{frame}$: Gray value of current frame; $\Delta F/F_{0\,\mathrm{correct}}$: Bleaching-corrected relative change in fluorescence; $F_0$: Resting fluorescence value; *e*: error. $\lambda$: wavelength; $K_I$: integral gain; i[k]: current sign of the control error. Equations 1–3 are based on the ImageJ Plugin "Correctbleach"[56].

When the OVC acquisition is complete, a results text file is given out, and the respective image stack is saved automatically as TIFF file. The results file comprises the selected parameters, the bleach correction function and its quality, the framerate, as well as time, (bleaching corrected-) raw gray- and $\Delta F/F_0$ values, and wavelength traces. In addition, status information (system on hold, adapting, or wavelength limits reached) is recorded for each time point of the clamp phase. For the "on-the-run" mode, an additional control window opens as soon as the measurement starts. Here, the holding $\Delta F/F_0$ values can be selected live, either via a range slider or with help of the keyboard arrow keys, while a live status update is displayed.

In addition, "pseudo I/V curve" software (Supplementary Code 1) is available to clamp relative fluorescence in 11 consecutive steps based on previously selected upper and lower limits, resulting in an output of the average $\Delta F/F_0$ value and associated wavelength achieved for each step (calculated as a mean of the last 25% of each step). An optical wavelength/$\Delta F/F_0$ curve is produced and can be translated into an estimated I/V-diagram using these linear calibration functions (Fig. 2J, L):

linear regression: $y = a + bx + e$, where *e* is assumed as i.i.d. residuals with mean 0.

Membrane potential as a function of $\Delta F/F_0$ holding (estimate: all observations, $n = 80$, w/o outliers)[a]

| | |
|---|---|
| Constant | −30.27*** |
| B | 2.25*** |
| $R^2$ | 0.8 |

linear regression: $y = a + bx + e$, where *e* is assumed as i.i.d. residuals with mean 0.

Current as a function of wavelength (estimate: all observations, $n = 72$)[a]

| | |
|---|---|
| Constant | 607.06*** |
| b | −1.21*** |
| $R^2$ | 0.8 |

[a]Estimate with robust (white heteroscedasticity consistent) standard errors.

***Significantly different from 0 at the 1% level.

A further software version involves a PID-controller to adjust the wavelength of the OVC (Supplementary Code 1). At each time point, the new wavelength is calculated by the addition of the output (u) of the PID equation to the calibration wavelength (Eqs. 7–9).

$$e = \Delta F/F_{0\,\mathrm{holding}} - \Delta F/F_0 \tag{7}$$

$$u = K_P * e + K_I * T_a * e_{sum} + K_D * \frac{e - e_{t-1}}{T_a} \qquad (8)$$

$$\lambda = \lambda_0 + u \qquad (9)$$

$e$: current error; $u$: output; $K_P$: proportional gain; $K_I$: integral gain; $K_D$: derivative gain; $e_{t-1}$: previous error; $T_a$: sampling rate; $e_{sum}$: summed up error; $\lambda_0$: calibration wavelength.

Parameter tuning was performed by applying the Ziegler–Nichols method to empirically tune the P, I, and D control gains for better control performance (Supplementary Fig. 2C). Therefore, $K_P$ was adapted until the system started to oscillate at the critical proportional gain ($K_{P\,crit}$). The oscillation period was noted as $T_{crit}$. PID coefficients were calculated according to Eqs. 10 and 11:

Ziegler–Nichols parameter tuning

| $K_P$ | $T_n$ | $T_v$ |
|---|---|---|
| **0.6 $K_{P\,crit}$** | 0.5 $T_{crit}$ | 0.12 $T_{crit}$ |

$K_P$: proportional gain; $K_{P\,crit}$: critical proportional gain; $T_{crit}$: period of oscillation; $T_n$: integration coefficient; $T_v$: derivation coefficient[36].

$$K_D = K_P * T_v \qquad (10)$$

$$K_I = K_P * \frac{1}{T_n} \qquad (11)$$

To also achieve a better control performance, a Kalman filter for sensor-signal smoothing was added (Supplementary Fig. 2A). Here, the error covariances of the measurement- and system noise of the Kalman filter were empirically adjusted so that the OVC controller dynamics achieved a desired performance. After initializing the system state with an a priori estimate (variables $\hat{x}_0$: estimated fluorescence at the beginning of the measurement and $P_0$: estimated error covariance of measured fluorescence vs. estimated fluorescence at the beginning of the measurement), prediction and correction are performed at each time step, alternately propagating the system state in time, and correcting it with new observations (Eqs. 12–17)[58]:

1. Prediction of the fluorescence value and the error covariance:

$$x^*(k+1|k) = \mathbf{A}\hat{x}(k) \qquad (12)$$

$$\mathbf{P}^*(k+1|k) = \mathbf{A}P(k)\mathbf{A}^\top + \mathbf{Q} \qquad (13)$$

$k$: discrete time step; A: system matrix, was here set to scalar value one (which is sufficient for error smoothing); $x^*$: predicted fluorescence; $P^*$: predicted error covariance of the estimate; Q: error covariance of the system noise; $P(k)$: estimated error covariance.

2. Calculation of the Kalman gain:

$$\mathbf{S}(k+1) = \mathbf{H}\mathbf{P}^*(k+1|k)\mathbf{H}^\top + \mathbf{R} \qquad (14)$$

$$\mathbf{K}(k+1) = \mathbf{P}^*(k+1|k)\mathbf{H}^\top \mathbf{S}^{-1}(k+1) \qquad (15)$$

S: auxiliary quantity for determining the Kalman gain; **H**: system output matrix; **R**: error covariance matrix of the measurement noise; **K**: Kalman gain.

3. Correction of the state estimate:

$$\hat{x}(k+1) = x^*(k+1|k) + \mathbf{K}(k+1)[z(k+1) - \mathbf{H}x^*(k+1|k)] \qquad (16)$$

$z(k+1)$: current measurement; $x^*(k+1,|,k)$: predicted measurement; $\hat{x}(k+1)$: current estimate.

4. Correction of the covariance estimation

$$\mathbf{P}(k+1) = [\mathbb{1} - \mathbf{K}(k+1)\mathbf{H}]\mathbf{P}^*(k+1|k) \qquad (17)$$

The system matrices $A$ and $H$ were set as constant scalars ($A = 1$, $H = 1$; constant system dynamics) since this assumption is perfectly sufficient for signal smoothing for the PID controller. Here, the sampling rate is fast enough that the Kalman filter can respond to membrane potential changes occurring in this time period with reasonable estimates.

Another software, "optical current clamp" (Supplementary Code 1) is used for the purely optical adaptation of a current clamp. The experimenter has the choice to set single pulses of certain wavelength and duration or to select step-like or continuous wavelength ramps prior to the experiment. Similar to the main OVC script (Supplementary Code 1), this software provides bleaching correction and live $\Delta F/F_0$ readout.

## Quality assessment of the OVC

The predictive accuracy for bleaching correction had $R^2$ values above 0.95 in most cases, and at least above 0.8 (Supplementary Fig. 6A). Experiments with lower $R^2$ values were attributable to APs occurring during the calibration phase (e.g., in pharyngeal muscle). As expected, the fraction of system saturation decreased once optogenetic tools for de- and hyperpolarization were simultaneously used, which stresses the importance of a second, opposing actuator (Supplementary Fig. 6B). While for the single-tool approaches, the percentage of system saturation was relatively high with $16.3 \pm 2.8\%$, this relation was significantly reduced for all bidirectional BiPOLES combinations. This was also reflected in the speed of the system, where compared to single-tool combinations, BiPOLES was significantly faster, particularly for driving excited or hyperpolarized states back towards resting potential (Supplementary Fig. 6C).

## Electrophysiology in C. elegans

Electrophysiological recordings from body wall muscle cells were conducted in immobilized and dissected adult worms as described previously[41]. Animals were immobilized with Histoacryl L glue (B. Braun Surgical, Spain) and a lateral incision was made to access neuromuscular junctions (NMJs) along the anterior ventral nerve cord. The basement membrane overlying body wall muscles was enzymatically removed by 0.5 mg/ml collagenase for 10 s (C5138, Sigma-Aldrich, Germany). Integrity of BWMs and nerve cord was visually examined via DIC microscopy. Recordings from BWMs were acquired in whole-cell patch-clamp mode at 20-22 °C using an EPC-10 amplifier equipped with Patchmaster software (HEKA, Germany). Voltage clamp experiments were conducted at the given holding potential (e.g., −24 mV). Membrane potentials in body wall muscle cells were recorded in current clamp mode. In regular recordings no additional current pulse was injected. For measurements of input resistance, a current pulse of −20 pA was injected via the Patchmaster software for 1000 ms in five consecutive pulses with 5 s breaks in between the current pulses.

The head stage was connected to a standard HEKA pipette holder for fire-polished borosilicate pipettes (1B100F-4, Worcester Polytechnic Institute, Worcester, MA, USA) of 4-10 MΩ resistance. The extracellular bath solution consisted of 150 mM NaCl, 5 mM KCl, 5 mM $CaCl_2$, 1 mM $MgCl_2$, 10 mM glucose, 5 mM sucrose, and 15 mM HEPES (pH 7.3 with NaOH, ≈330 mOsm). The internal/patch pipette solution consisted of K-gluconate 115 mM, KCl 25 mM, $CaCl_2$ 0.1 mM, $MgCl_2$

5 mM, BAPTA 1 mM, HEPES 10 mM, $Na_2ATP$ 5 mM, $Na_2GTP$ 0.5 mM, cAMP 0.5 mM, and cGMP 0.5 mM (pH 7.2 with KOH, ≈ 320 mOsm). For some experiments, BAPTA was replaced by EGTA (5 mM).

For light activation a monochromator (Polychrome V, Thermo Scientific) was used, ranging from 400 to 600 nm at 300 µW/mm². To create conditions as similar as possible to those for the OVC experiments, additional excitation was performed with a laser at 637 nm. Subsequent analysis and graphing were performed using Patchmaster, and Origin (Originlabs).

Electrophysiological recordings from the DVB neuron were conducted in immobilized and dissected adult worms as described previously[46] with minor modifications. Both dissection and recording were performed at room temperature. Briefly, an animal was immobilized with cyanoacrylate adhesive (Vetbond tissue adhesive; 3 M) along the ventral side of the posterior body on a Sylgard 184-coated (Dow Corning) glass coverslip. A small longitudinal incision was made using a diamond dissecting blade (Type M-DL 72029 L; EMS) in the tail region along the glue line. The cuticle flap was folded back and glued to the coverslip with GLUture Topical Adhesive (Abbott Laboratories), exposing DVB. The coverslip with the dissected preparation was then placed into a custom-made open recording chamber (-1.5 ml volume) and treated with 1 mg/ml collagenase (type IV; Sigma) for -10 s by hand pipetting. The recording chamber was subsequently perfused with the standard extracellular solution using a custom-made gravity-feed perfusion system for -10 ml. The standard pipette solution was (all concentrations in mM): K-gluconate 115; KCl 15; KOH 10; $MgCl_2$ 5; $CaCl_2$ 0.1; $Na_2ATP$ 5; NaGTP 0.5; Na-cGMP 0.5; cAMP 0.5; BAPTA 1; Hepes 10; Sucrose 50, with pH adjusted to 7.2, using KOH, osmolarity 320–330 mOsm. The standard extracellular solution was: NaCl 140; NaOH 5; KCl 5; $CaCl_2$ 2; $MgCl_2$ 5; Sucrose 15; Hepes 15; Dextrose 25, with pH adjusted to 7.3, using NaOH, osmolarity 330–340 mOsm. Liquid junction potentials were calculated and corrected before recording. Electrodes with resistance of 20–30 MΩ were made from borosilicate glass (BF100-58-10; Sutter Instruments) using a laser pipette puller (P-2000; Sutter Instruments) and fire-polished with a microforge (MF-830; Narishige). Recordings were performed using single-electrode whole-cell current-clamp (Heka, EPC-10 USB) with two-stage capacitive compensation optimized at rest, and series resistance compensated to 50%.

## Preparation of organotypic hippocampal slice cultures and transgene delivery

All procedures were in agreement with the German national animal care guidelines and approved by the independent Hamburg state authority for animal welfare (Behörde für Justiz und Verbraucherschutz). They were performed in accordance with the guidelines of the German Animal Protection Law and the animal welfare officer of the University Medical Center Hamburg-Eppendorf.

Organotypic hippocampal slices were prepared from Wistar rats of both sexes (Jackson-No. 031628) at post-natal days 5–7 as previously described[59]. In brief, dissected hippocampi were cut into 350 µm slices with a tissue chopper and placed on a porous membrane (Millicell CM, Millipore). Cultures were maintained at 37 °C, 5% $CO_2$ in a medium containing 80% MEM (Sigma M7278), 20% heat-inactivated horse serum (Sigma H1138) supplemented with 1 mM L-glutamine, 0.00125% ascorbic acid, 0.01 mg ml⁻¹ insulin, 1.44 mM $CaCl_2$, 2 mM $MgSO_4$ and 13 mM D-glucose. No antibiotics were added to the culture medium. Pre-warmed medium was replaced twice per week.

For transgene delivery, organotypic slice cultures were transduced at DIV 3–5 with a recombinant adeno-associated viral vector (rAAV) encoding a soma-targeted version of BiPOLES[27] (AAV9-CaMKII-somBiPOLES-mCerulean, Addgene #154948). The rAAV was locally injected into the CA1 region using a Picospritzer (Parker, Hannafin) by a pressurized air pulse (2 bar, 100 ms) expelling the viral suspension into the slice[60]. During virus transduction, membranes carrying the slices were kept on pre-warmed HEPES-buffered solution. At DIV 14–16,

individual CA1 pyramidal cells in the previously somBiPOLES-transduced slices were transfected by single-cell electroporation[61] with the plasmid pAAV-QuasAr2-ts-ER at a final concentration of 10 ng/µl in K-gluconate-based solution consisting of (in mM): 135 K-gluconate, 10 HEPES, 4 $Na_2$-ATP, 0.4 Na-GTP, 4 $MgCl_2$, 3 ascorbate, 10 $Na_2$-phosphocreatine (pH 7.2). An Axoporator 800 A (Molecular Devices) was used to deliver 50 hyperpolarizing pulses (−12 V, 0.5 ms) at 50 Hz. During electroporation slices were maintained in pre-warmed (37 °C) HEPES-buffered solution (in mM): 145 NaCl, 10 HEPES, 25 D-glucose, 2.5 KCl, 1 $MgCl_2$ and 2 $CaCl_2$ (pH 7.4, sterile filtered). A plasmid encoding hSyn-mCerulean (at 50 ng/µl) was co-electroporated and served as a marker to rapidly identify cells putatively co-expressing QuasAr2 and somBiPOLES under epifluorescence excitation.

## OVC voltage imaging and electrophysiology experiments in rat organotypic slices

At DIV 19-21, voltage imaging and/or whole-cell patch-clamp recordings of transfected CA1 pyramidal neurons were performed. Experiments were done at room temperature (21–23 °C) using a BX51WI upright microscope (Olympus) equipped with an EMCCD camera (Evolve 512 Delta, Photometrics), dodt-gradient contrast and a Double IPA integrated patch amplifier controlled with SutterPatch software (Sutter Instrument). QuasAr2 was excited with a 637 nm red laser (OBIS FP 637LX, Coherent) at 0.52 W/mm² via a dichroic mirror (660 nm, Edmund Optics) and voltage-dependent fluorescence was detected through the objective LUMPLFLN 60XW (Olympus) and through an emission filter (732/68 nm Bandpass filter, Edmund Optics). somBiPOLES was activated using a monochromator (Polychrome V, Till Photonics/Thermo Scientific), set to emit light from 400 to 600 nm at 1.7 mW/mm². Laser and monochromator light were combined using a 605 nm dichroic mirror (Edmund Optics). Irradiance was measured in the object plane with a PM400 power meter equipped with a S130C photodiode sensor (Thorlabs) and divided by the illuminated field of the Olympus LUMPLFLN 60XW objective (0.134 mm²). As for *C.elegans*, OVC experiments were performed at ≈ 90 fps with 10 ms exposure and a binning of 4 × 4.

For electrophysiology, patch pipettes with a tip resistance of 3-4 MOhm were filled with intracellular solution consisting of (in mM): 135 K-gluconate, 4 $MgCl_2$, 4 $Na_2$-ATP, 0.4 Na-GTP, 10 $Na_2$-phosphocreatine, 3 ascorbate, 0.2 EGTA, and 10 HEPES (pH 7.2). Artificial cerebrospinal fluid (ACSF) consisted of (in mM): 135 NaCl, 2.5 KCl, 2 $CaCl_2$, 1 $MgCl_2$, 10 Na-HEPES, 12.5 D-glucose, 1.25 $NaH_2PO_4$ (pH 7.4). Synaptic transmission was blocked by adding 10 µM CPPene, 10 µM NBQX, and 100 µM picrotoxin (Tocris) to the extracellular solution. Measurements were corrected for a liquid junction potential of −14,5 mV. Access resistance of the recorded neurons was continuously monitored and recordings above 30 MΩ were discarded.

## Statistical methods

Statistical methods used are described in the figure legends. No sample was measured multiple times, and no samples were excluded from analysis. Statistical analysis was performed in Prism (Version 8.0.2, GraphPad Software, Inc.), OriginPro 2021 (OriginLab Corporation, Northampton, USA), Microsoft Excel 2016, 2019, SutterPatch V2 (Sutter Instruments), MATLAB 2016b, 2019b (Mathworks), Stata 12 ic, or ImageJ v1.53c. No statistical methods were applied to predetermine sample size. However, sample sizes reported here are consistent to data presented in previous publications, and which typically allow distinction of significant differences in the types of experiments used. Throughout, n indicates biological replicates (i.e., animals analyzed), with additional mention of specific data that was analyzed (e.g., action potential number), where appropriate. Data was tested for normality prior to statistical inference. Data is given as means ± SEM, if not otherwise stated. Significance between data sets after paired or two-sided $t$ test or ANOVA, with Bonferroni correction for multiple comparisons, is given as $p$ value (*$p \le 0.05$; **$p \le 0.01$; ***$p \le 0.001$), when not otherwise stated. Group

data are presented as box plots with median (line), 25th–75th quartiles (box), mean (open dot) and whiskers, representing 1.5× inter quartile range (IQR). Single data points are shown for $n \leq 10$, or for outliers. The $R^2$ coefficient of determination was determined as: $R^2 = 1 - SSE/SSD$ (SSE = sum of the squared errors; SSD = sum of the squared deviations about the mean). The linear regressions were estimated with robust standard errors (White heteroscedasticity consistent).

## Reporting summary

Further information on research design is available in the Nature Portfolio Reporting Summary linked to this article.

## Data availability

Original data (videos) are available from the authors on request. Source data are provided with this paper.

## Code availability

The software/code used for operating the OVC is written in Beanshell, the scripting language used by µManager freeware. Scripts ("OVC_main_script", "OVC_on_the_run_script", "OVC_4_step_script", "OVC_pseudo-IV_script", "optical_current_clamp_script") are available as a zip archive in Supplementary Code 1.

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

## Acknowledgements

We thank Jiajie Shao for support in confocal microscopy, Adam Cohen for reagents, and Sven Plath, Regina Wagner, Hans-Werner Müller, Franziska Baumbach, Nico Sturman, Adrian Breicher, Timotheus Kozlowski, Rolf Bergs, and Kathrin Sauter for expert technical assistance. We are indebted to Mathias Pasche, Wagner Steuer Costa, Achilleas Frangakis, and Ernst Stelzer for advice. We acknowledge funding by the following organizations: Goethe University (A.G.); Deutsche Forschungsgemeinschaft, SFB807/B02 (A.G.), SPP1926-XIb/WI 4485/3-2 (J.S.W.), and SFB1315/C01 (J.V., P.H.); Max Planck Society, PhD fellowship (A.C.F.B.); Chan Zuckerberg Initiative (Q.L., C.I.B.).

## Author contributions

Conceptualization: A.C.F.B., A.G., J.S.W. Methodology: A.C.F.B., A.B., J.V., P.H., J.S.W. Investigation: A.C.F.B., J.F.L., S.R.-R., Q.L., A.N., C.W., N.Z., H. Durmaz, H. Dill, M.J. Visualization: A.C.F.B., A.G. Funding acquisition: A.G., A.C.F.B., J.S.W., P.H., C.I.B. Project administration: A.G. Supervision: A.G., J.S.W., C.I.B. Writing—original draft: A.C.F.B., A.G. Writing—review and editing: A.C.F.B., A.G., S.R.-R., J.S.W., J.V., C.I.B., P.H.

## Funding

## Competing interests

A patent application has been filed: Applicants: A.C.F.B., A.G., Goethe University; inventors: Amelie Bergs, Alexander Gottschalk; application number: EP 21162331.9; PCT/EP2022/056030, pending, claims focusing on use of an optogenetic voltage clamp for providing models of neurodegenerative or cardiac diseases and for functional characterization or screening of ion channels and/or drugs that affect ion channels. The remaining authors declare no competing interests.
