## [Peer Review File · Nature Communications]

All-optical closed-loop voltage clamp for precise control of muscles and neurons in live animalsEditorial Note: This manuscript has been previously reviewed at another journal that is not operating a transparent peer review scheme. This document only contains reviewer comments and rebuttal letters for versions considered at Nature Communications.

Reviewers' Comments:

Reviewer #1:

Remarks to the Author:

This paper describes an "Optical Voltage Clamp" based on bidirectional optogenetic control and a far-red voltage indicator. Clearly a lot of work went into the system and it is interesting to see how far the authors pushed the technology. As in my original assessment, the overall performance is quite limited in dynamic range and bandwidth, so the system is likely to be limited in use to a subset of *C. elegans* researchers. The overall impact of the work is modest. Nonetheless, the system is worth publishing, either in Nature Communications or a more specialized journal, as others may build on the work (or avoid trying similar approaches after seeing the difficulties).

Unfortunately the authors' replies to the previous critiques were largely nonresponsive. I will try once more.

The abstract says that the OVC can "dynamically clamp spiking" after discussing the application in rat neurons. This is misleading, because it can only do this in *C. elegans*, not in rat neurons. The authors should revise to say that the OVC can "dynamically clamp spiking in *C. elegans*"

Line 85, 436: The authors claim that the feedback loop functions at 100 Hz, but there is no evidence to support that claim. To support the claim, the authors should show that a change in fluorescence can be imposed and stabilized in 10 ms, not just that the software loop runs at 100 Hz. The true bandwidth is closer to 10 Hz (148 ms or 88 ms). The fact that the software update rate is 100 Hz is irrelevant to the true system bandwidth. Given the fact that the system is already substantially slower than the 100 Hz update rate, the authors would need to better justify their claim that increasing the measurement bandwidth to 1 kHz would actually speed the whole system.

Letters below refer to comments in previous round of review:

(a) I previously asked for a characterization of the accuracy of the system. The authors' response that 93% of the time there is zero deviation is, of course, nonsensical. No control system is perfect, and claiming that this one is raises doubt about the overall seriousness of the effort. The control system might be good, but to establish how good it is, the authors should actually quantify the power spectrum of the deviations, i.e. calculate a frequency-dependent power spectral density of $(DF/F)_{\text{measured}} - (DF/F)_{\text{target}}$. The authors already have the data so this is minimal additional work—I do not understand why they are resisting doing this.

The authors state that the tolerance range of their feedback system is $\pm 1\%$ DF/F. This seems to be the closest statement they have made to establishing an accuracy of their system. This might be the authors' "desired precision", but a user would still like to know how much precision is attainable. This tolerance range should be more clearly and prominently stated in the main text. The tolerance is a substantial fraction of the total dynamic range ($\pm 5\%$).

(b) I previously raised the question of how much time it takes to detect enough photons to achieve a given shot noise-limited precision, and the authors replied that this must depend on the imaging

system, camera, and biology which a Reviewer could not possibly know. Unfortunately, the authors are wrong. The shot noise-limited accuracy in DF/F is equal to $1/\sqrt{N}$, where N is the number of *detected* photons. This is a hard physical limit, irrespective of biology, optics, camera—one can always do worse than this, but never better. It would be nice to know how close the system is to the shot-noise limit.

(d) Since this Reviewer misunderstood how the system works, perhaps the authors could do a better job explaining it. The manuscript should clearly explain what limits the feedback bandwidth (it seems that the monochromator is set to take small steps in order to maintain stability of the feedback loop, so the large-step response time is many-fold slower than the loop time).

Reviewer #1 (Remarks to the Author):

our answers

This paper describes an “Optical Voltage Clamp” based on bidirectional optogenetic control and a far-red voltage indicator. Clearly a lot of work went into the system and it is interesting to see how far the authors pushed the technology. As in my original assessment, the overall performance is quite limited in dynamic range and bandwidth, so the system is likely to be limited in use to a subset of *C. elegans* researchers. The overall impact of the work is modest. Nonetheless, the system is worth publishing, either in Nature Communications or a more specialized journal, as others may build on the work (or avoid trying similar approaches after seeing the difficulties). Unfortunately the authors’ replies to the previous critiques were largely nonresponsive. I will try once more.

(1) The abstract says that the OVC can “dynamically clamp spiking” after discussing the application in rat neurons. This is misleading, because it can only do this in *C. elegans*, not in rat neurons. The authors should revise to say that the OVC can “dynamically clamp spiking in *C. elegans*”

We agree and have changed the manuscript accordingly.

(2) Line 85, 436: The authors claim that the feedback loop functions at 100 Hz, but there is no evidence to support that claim. To support the claim, the authors should show that a change in fluorescence can be imposed and stabilized in 10 ms, not just that the software loop runs at 100 Hz. The true bandwidth is closer to 10 Hz (148 ms or 88 ms). The fact that the software update rate is 100 Hz is irrelevant to the true system bandwidth. Given the fact that the system is already substantially slower than the 100 Hz update rate, the authors would need to better justify their claim that increasing the measurement bandwidth to 1 kHz would actually speed the whole system.

We agree that it is important to further evaluate the system bandwidth and speed limits and that “feedback loop functions at 100 Hz” is ambiguous. To clarify, the 100 Hz frequency corresponds to the sampling rate (f_{sample}) of our feedback controller (rate by which the processor calculates and sets a new output value (wavelength)), whereas intervals of 88 ms and 148 ms refer to the time required to reach the predetermined $\pm 1\%$ (tolerance range) of its target value (defined as ‘transition time’, **line 185 / revised version: lines 145-146**). Moreover, we agree and are aware that a controller with a sampling rate of 100 Hz cannot reach the setpoint stably within 10 ms (one sampling step), because compensation of controller deviation always requires a multiple of the sampling time. The duration of individual control events is comparably long when large transitions (-5% or $+10\% \Delta F/F_0$) are imposed, yet it is much faster when small (intrinsic) deviations from the tolerance range are counteracted. To extend the characterization of our time-dependent closed loop control system beyond transition times (i.e., the system counteracting small deflections from the tolerance range), we evaluated all control events (**Supplementary Fig. 6F**): Almost 50 % of those events were completed ($\Delta F/F_0$ returning back in the tolerance range) within 20-30 ms.

Further, the OVC’s transition time depends on the sampling rate: An initial test series at 40 Hz sampling rate resulted in ca. 3-fold longer transition times for imposed steps (**Supplementary Fig. 6E**). In general, a higher sampling rate can shorten the transition time, since the controller now applies many more control signals (wavelength changes) to the system within a given period. Therefore, closed loop transition times do not allow to approximate the true open loop system frequency. To this end, we added data on the open loop step response of our system (simply providing de- or hyperpolarizing stimuli under same conditions, **Supplementary Fig. 1D**): In this case, the system required barely 20 ms to reach $\pm 5\% \Delta F/F_0$, which we conclude to be an approximation of the system’s time constant. In this context, sampling rates should exceed the open loop system frequency by at least a factor greater than 2, but ideally by 6-20 (theoretical sampling bandwidth of 300-1000 Hz; Lunze, Regelungstechnik 2, 9th ed., p. 410; Shannon, doi: 10.1109/jrproc.1949.232969). Hence, identification of the optimal sampling rate could bring the transition time closer to the system’s time constant and optimize control quality. The optimal sampling rate is not exclusively dependent on the system frequency, but is also limited by the necessary computing time of the processor and the photon count at higher rates (see **comment 4**).

All in all, we agree that the speed of the closed loop system at 100 Hz sampling rate is mainly defined by the transition times, which we highlighted more prominently in the discussion (**line 404-405**). We have also added a discussion on estimated optimal sampling rates and maximum system speed to the manuscript (**lines 114-117, 405-417**) and clarified conceptual ambiguities, especially in **lines 85 and 436 (now: 83-84 and 361)**.

Letters below refer to comments in previous round of review:

(3) (a) I previously asked for a characterization of the accuracy of the system. The authors’ response that 93% of the time there is zero deviation is, of course, nonsensical. No control system is perfect, and claiming that this one is raises doubt about the overall seriousness of the effort. The control system might be good, but to establish how good it is, the authors should actually quantify the power spectrum of the deviations, i.e. calculate a

frequency-dependent power spectral density of $(DF/F)_{\text{measured}} - (DF/F)_{\text{target}}$. The authors already have the data so this is minimal additional work—I do not understand why they are resisting doing this. The authors state that the tolerance range of their feedback system is $\pm 1\%$ DF/F . This seems to be the closest statement they have made to establishing an accuracy of their system. This might be the authors' "desired precision", but a user would still like to know how much precision is attainable. This tolerance range should be more clearly and prominently stated in the main text. The tolerance is a substantial fraction of the total dynamic range ($\pm 5\%$).

Original comment: A basic characterization of a feedback system should include time-dependent accuracy. One would like to know in a given single-trial averaging window, the r.m.s. deviation between target and measured DF/F ; or equivalently, the power spectrum of the deviations, i.e. of $(DF/F)_{\text{measured}} - (DF/F)_{\text{target}}$. These quantities probably depend on baseline voltage because at depolarized conditions the system is working to combat action potentials.

We thank the reviewer for clarifying this previous criticism. For the first part, we think our description was misleading: The 93 % referred to the time of $\Delta F/F_0$ being within the $\pm 1\%$ tolerance range, not to the time of zero deviation from the target value. The tolerance range is not a post-evaluation criterion, but a selectable range that is dictated to the system before each measurement. Once the tolerance range is reached, no further adaptation is performed (see method section, line 561-562), i.e. the system does not attempt to move the fluorescence value further towards the exact clamp value. Thus, the selectable tolerance range corresponds to the hysteresis of the controller, i.e. it is the range in which the actual value may fluctuate, to increase stability. We adapted the text to better explain this mode of operation of the controller (lines 125-128).

For the second part, we now provide a more detailed analysis that discriminates the control deviation (difference of actual $\Delta F/F_0$ from target value) also within the tolerance limits (frequency distribution / histogram, Supplementary Fig. 6G), although this is only of limited value, as is evident from the above-mentioned reasons. This showed that 50 % of the time, the deviations are less than 0.5 %, 84 % of the time below 1 % and 96 % of the time below 2 %. Furthermore, we provide the requested time-dependent measure of accuracy, i.e. the r.m.s. deviation of measured from target $\Delta F/F_0$ (Supplementary Fig.6H), as well as the power spectral density of the same difference (rebuttal letter Fig. 1).

Figure 1. Power spectral density of the control deviation. (A) Full time series. **(B)** PSD representation of closed loop step responses (blue) and associated control deviations (red), that were isolated, summed up, derived, and normalized to generate the system's impulse response.

(4) (b) I previously raised the question of how much time it takes to detect enough photons to achieve a given shot noise-limited precision, and the authors replied that this must depend on the imaging system, camera, and biology which a Reviewer could not possibly know. Unfortunately, the authors are wrong. The shot noise-limited accuracy in DF/F is equal to $1/\sqrt{N}$, where N is the number of *detected* photons. This is a hard physical limit, irrespective of biology, optics, camera—one can always do worse than this, but never better. It would be nice to know how close the system is to the shot-noise limit.

Original comment: To get a 1% accuracy on DF/F requires at least 10^4 photons from the sample, and 0.1% requires 10^6 photons. It takes time to get these photons, and the QuasAr reporters are quite dim. The paper does not provide any quantitative estimates of photon counts or of the shot-noise floor.

This was a misunderstanding. We assumed that the reviewer implied that the voltage sensors were too dim, without knowing the exact photon count. We thank the reviewer for his advice to discuss the shot noise-limited precision of our system. As requested, we have added the listed specifications to the manuscript (**Supplementary table 1**): In general, for the CMOS camera we use (Kinetix22), read noise ($1.6e^-$) and dark current ($1.27 e^-/p/sec$) are negligible, that is why these camera sensors are described as shot-noise limited by the manufacturer. As the reviewer explained, the nature of light imposes a theoretical limit on the achievable SNR for a given photon flux. With our standard frame rate of 100 fps (10 ms exposure), based on the electron count, we estimate an average signal of ca. 1,800,000 photons per frame (the OVC averages grey values of a ROI with ca. 350 pixels per timepoint, i.e., accuracy is based on photons captured in 350 pixels at each timepoint). Hence, the shot-noise floor is 0.08 % of the total signal power.

(5) (d) Since this Reviewer misunderstood how the system works, perhaps the authors could do a better job explaining it. The manuscript should clearly explain what limits the feedback bandwidth (it seems that the monochromator is set to take small steps in order to maintain stability of the feedback loop, so the large-step response time is many-fold slower than the loop time).

As described in **(2)**, we distinguish

- the sampling rate of the I-controller (which is limited by the software to 100 Hz),
- the open loop system frequency based on the step response analysis,
- the time needed to achieve clamping stability for the preset clamping values (transition times in closed loop).

The latter depends on the sampling rate and is limited downwards by the system frequency.

Wavelength adaptation occurs via an implemented I-controller (described in the method section, **lines 552-554, 561-562**). Hence, the wavelength is changed with respect to the deviation size (**Eq. 6**), i.e., the larger the control deviation, the larger is the wavelength change. The integral gain is selected jointly with the tolerance range such that an appropriate stability is achieved. Thus, the monochromator is not statically set to take small steps, but reacts dynamically depending on the extent of deviation from the target $\Delta F/F_0$ value. Nevertheless, for large control deviations (5% transitions), the controller runs through more consecutive steps (large adjustments at the beginning, followed by fine tuning when the deviation becomes smaller), while smaller deviations are compensated within few consecutive frames. As stated above **(2)**, the system's sampling rate is currently limited by the software to 100 Hz, which in turn limits the closed loop speed of the system defined by its transition times. In the manuscript, we have expanded our discussion regarding our estimates on the maximally possible system speed (**lines 114-117, 404-417**).

Reviewers' Comments:

Reviewer #1:

Remarks to the Author:

The authors and I have reached a fixed point in the review feedback loop. I am happy for it to be published as is.